# Beyond the Heatmap: A Rigorous Evaluation of Component Impact in MCTS-Based TSP Solvers

**Xuanhao Pan**[1,†]    **Chenguang Wang**[1,†]    **Chaolong Ying**[1]    **Ye Xue**[1,2,3]    **Tianshu Yu**[1*]

[1]School of Data Science, The Chinese University of Hong Kong, Shenzhen
[2]Shenzhen Research Institute of Big Data
[3]School of Intelligent Systems Engineering, Shenzhen Campus of Sun Yat-sen University
Shenzhen, China
{xuanhaopan, chenguangwang, chaolongying}@link.cuhk.edu.cn
xuey57@mail.sysu.edu.cn   yutianshu@cuhk.edu.cn
[†]Equal contribution

## Abstract

The "Heatmap + Monte Carlo Tree Search (MCTS)" paradigm has recently emerged as a prominent framework for solving the Traveling Salesman Problem (TSP). While considerable effort has been devoted to enhancing heatmap sophistication through advanced learning models, this paper rigorously examines whether this emphasis is justified, assessing the relative impact of heatmap complexity versus MCTS configuration. Our extensive empirical analysis across diverse TSP scales, distributions, and benchmarks reveals two pivotal insights: **1**) The configuration of MCTS strategies strongly influences solution quality, underscoring the importance of systematic tuning to achieve optimal results and enabling valid comparisons among different heatmap methodologies. **2**) A rudimentary, parameter-free heatmap based on the intrinsic $k$-nearest neighbor structure of TSP instances, when coupled with an optimally tuned MCTS, can match or surpass the performance of more sophisticated, learned heatmaps, demonstrating robust generalizability on problem scale and distribution shifts. To facilitate rigorous and fair evaluations in future research, we introduce a streamlined pipeline for standardized MCTS hyper-parameter tuning. Collectively, these findings challenge the prevalent assumption that heatmap complexity is the primary determinant of performance, advocating instead for a balanced integration and comprehensive evaluation of both learning and search components within this paradigm. Our code is available at: `https://github.com/LOGO-CUHKSZ/beyond-heatmap-mcts-tsp`.

## 1 Introduction

The Traveling Salesman Problem (TSP) remains a fundamental challenge in combinatorial optimization, drawing considerable interest from theoretical and applied research communities. As an NP-hard problem, the TSP serves as a crucial benchmark for evaluating novel algorithmic strategies for finding optimal or near-optimal solutions efficiently (Applegate et al., 2009). Its practical significance spans logistics, transportation, manufacturing, and telecommunications, where efficient routing is paramount for cost minimization and operational improvement (Helsgaun, 2017; Nagata and Kobayashi, 2013). Recent machine learning advancements have spurred new methodologies for tackling TSP, notably the "Heatmap + Monte Carlo Tree Search (MCTS)" paradigm (Fu et al., 2021). Leveraging learned heatmaps to guide MCTS in refining solutions, this approach has demonstrated success on large-scale instances and inspired a proliferation of methods (Qiu et al., 2022; Sun and Yang, 2023; Min et al., 2024). This rapid development signals a maturing field where systematic evaluation, comparison, and validation of emerging techniques are increasingly essential.

Within this "Heatmap + MCTS" framework, a primary research thrust has centered on enhancing heatmap generation, often through increasingly sophisticated learning models, from supervised learning (Fu et al., 2021) to diffusion models (Sun and Yang, 2023). The underlying assumption is

---

*Corresponding author: `yutianshu@cuhk.edu.cn`

often that heatmap sophistication directly translates to superior solution quality. But is this pursuit of complexity the only, or even optimal, path to performance gains? Has the impact of MCTS configurations—the search component responsible for translating heatmap guidance into concrete solutions—been fully acknowledged and systematically investigated? Although numerous solvers have emerged claiming performance improvements, there remains a lack of evaluation-centered scrutiny regarding the actual influence of the MCTS component and the true necessity of intricate heatmap designs. Our work aims to address this gap, challenging the potential cognitive bias that "more complex heatmaps consistently lead to better performance" and providing clarity for researchers and practitioners.

This work presents a rigorous, evaluation-centered analysis of the "Heatmap + MCTS" paradigm for TSP. Our primary objective is to examine the deep impact of MCTS configurations and re-evaluate the necessity of heatmap complexity. The central argument of our evaluation is twofold: *first*, that strategic MCTS calibration substantially influences solution quality, demanding meticulous attention; and *second*, that our proposed GT-Prior—a simple, parameter-free $k$-nearest neighbor heatmap—can rival or even surpass complex learned heatmaps while also demonstrating strong generalization ability. Our evaluation spans various heatmap generation methods, from sophisticated learning-based models to this GT-Prior, and scrutinizes diverse MCTS hyperparameter settings. The empirical validation is performed on TSP instances of varying scales (TSP-500, TSP-1000, and TSP-10000), covering diverse problem structures through various synthetic distributions (including uniform, clustered, explosion, and implosion) and established real-world TSPLIB benchmarks.

The novelty of this paper lies not in proposing a new state-of-the-art solver, but in the rigor of its evaluation process and the critical insights derived. Our contributions are primarily evaluative:

- We empirically quantify and thereby reveal the often-underestimated significance of MCTS configurations in optimizing TSP solutions. Fine-tuning MCTS parameters such as exploration constant and node expansion criteria demonstrably impacts solution quality, urging a re-prioritization in algorithm design.
- We challenge the prevailing emphasis on heatmap complexity by demonstrating that a simple, parameter-free heatmap grounded in the $k$-nearest neighbor nature of TSP (termed GT-Prior in this work) exhibits strong performance and generalizability across diverse problem scales when combined with an optimized MCTS. This baseline serves to assess the added value of more intricate heatmap models.
- We introduce a streamlined MCTS hyperparameter tuning pipeline, offering a practical tool to facilitate fairer and more robust comparisons in future research on heatmap designs.

These findings collectively advocate for a more holistic understanding and balanced integration of learning and search components within the "Heatmap + MCTS" paradigm. Our work seeks to guide future research towards frameworks that synergistically harness both components, leading to more efficient, robust, and practically deployable TSP solvers, all while aligning with the foundational motivation of this research line: *to better solve large-scale TSP by any means*.

## 2 HEATMAP + MCTS: BACKGROUND AND CURRENT PERSPECTIVES

This section outlines the foundations of the "Heatmap + Monte Carlo Tree Search" paradigm for solving the Traveling Salesman Problem. We formalize the TSP and its heatmap representation, describe the adapted MCTS framework, review key methodological developments, and examine the current perspectives that motivate our evaluation.

### 2.1 TRAVELING SALESMAN PROBLEM DEFINITION

The Traveling Salesman Problem (TSP) is a classic combinatorial optimization problem defined over a set of $N$ points $I = \{(x_i, y_i)\}_{i=1}^N$ in the Euclidean plane, where each point denotes a city located at coordinates $(x_i, y_i) \in [0,1]^2$. The Euclidean distance between any two cities $i$ and $j$ is calculated by $d_{ij} = \sqrt{(x_i - x_j)^2 + (y_i - y_j)^2}$. The objective is to find the shortest closed tour that visits each city exactly once. This optimal tour is represented as a permutation $\pi^* = (\pi_1^*, \pi_2^*, \ldots, \pi_N^*)$, minimizing

the total length:

$$L(\pi^*) = \sum_{i=1}^{N-1} d_{\pi_i^* \pi_{i+1}^*} + d_{\pi_N^* \pi_1^*}. \tag{1}$$

The performance of a feasible solution $\pi$ is measured using the optimality gap:

$$Gap = \left( \frac{L(\pi)}{L(\pi^*)} - 1 \right) \times 100\%. \tag{2}$$

In the "Heatmap + MCTS" paradigm, the solution process is guided by a heatmap $P^N \in [0, 1]^{N \times N}$, where each entry $P_{ij}^N$ represents the estimated probability that edge $(i, j)$ appears in the optimal tour. This heatmap serves as a probabilistic prior that informs the subsequent search process.

## 2.2 The Monte Carlo Tree Search Framework for TSP

Originally introduced by Fu et al. (2021) to integrate learned heatmap priors with Monte Carlo search, this MCTS framework has become the de facto search backbone for heatmap-guided TSP solvers. MCTS formulates the TSP as a Markov Decision Process (MDP), where each state represents a valid tour and actions correspond to $k$-opt moves modifying the current solution. While many follow-up studies have reused its core procedure with only superficial adjustments, few have explored deeper search-centric refinements.

In this framework, the search begins by constructing an initial tour: edges are sampled with probability proportional to $e^{P_{ij}^N}$, where $P^N$ is the heatmap prior. The edge weight matrix $W$ is initialized as $W_{ij} = 100 \cdot P_{ij}^N$, the access frequency matrix $Q$ is set to zero, and the overall move counter $M$ starts at zero. Each node in the TSP graph maintains a candidate set to constrain future edge selections. During each simulation, a set of $k$-opt moves is generated and evaluated using the potential function in Equation (3), guiding the search toward higher-quality tours through repeated updates and restarts.

$$Z_{ij} = \frac{W_{ij}}{\Omega_i} + \alpha \sqrt{\frac{\ln(M+1)}{Q_{ij}+1}}, \tag{3}$$

where $\Omega_i = \sum_{j \neq i} W_{ij}$ normalizes the edge weights from node $i$, and $\alpha$ is an exploration coefficient. Edges with higher potential are more likely to be selected.

If a generated move yields a shorter tour ($\Delta L < 0$), it is accepted and applied. Otherwise, the search restarts from a newly sampled initial tour. After each move, MCTS updates the edge weights to reflect the observed improvement:

$$W_{ij} \leftarrow W_{ij} + \beta \left( \exp \left( \frac{L(\pi) - L(\pi')}{L(\pi)} \right) - 1 \right), \tag{4}$$

where $\beta$ is the learning rate. The access matrix $Q$ is incremented for all modified edges. The process iterates until a fixed time limit is reached, at which point the best tour encountered is returned.

## 2.3 Evolution and Prevailing Research in Heatmap-Guided MCTS

The "Heatmap + MCTS" framework, introduced by Fu et al. (2021), marked a significant shift in TSP research by pairing neural heatmap predictions with Monte Carlo Tree Search. Their method used attention-based GCNs to estimate edge probabilities, which then guided a stochastic search to build high-quality tours. This design has inspired numerous variants focused on refining heatmap quality.

Subsequent efforts introduced more sophisticated models to enhance generalization and structure awareness. DIMES employed meta-learned GNNs (Qiu et al., 2022); DIFUSCO leveraged diffusion-based generative models (Sun and Yang, 2023); and UTSP proposed an unsupervised learning strategy (Min et al., 2024). More recently, SoftDist (Xia et al., 2024) explored a simpler, distance-based heatmap, reflecting growing skepticism toward model complexity.

However, while heatmap design has seen continuous innovation, the search component—MCTS—has received comparatively less attention. Most prior works adopt default configurations with minimal tuning, and few report the impact of auxiliary steps such as sparsification or additional supervision. As a result, the actual contribution of MCTS to overall performance remains under-investigated.

## 2.4 CURRENT PERSPECTIVES AND POTENTIAL OVERSIGHTS

This disparity in research focus reflects several implicit views that have shaped the paradigm: **1**) that heatmap complexity is the primary driver of performance, justifying the emphasis on model sophistication; **2**) that default or minimally tuned MCTS configurations are sufficient for fair comparison, suggesting the search process is either secondary or robust by design; **3**) that MCTS itself is well understood, with its impact assumed to be stable across different problem scales and heatmap types.

We challenge these views through systematic evaluation. Our results show that MCTS tuning plays a pivotal role—often matching or exceeding the effect of heatmap refinement—and that a simple, parameter-free prior can outperform complex models when coupled with optimized search. These findings argue for a more balanced and transparent evaluation framework in future work.

## 3 EVALUATION METHODOLOGY

This section outlines our experimental framework for a rigorous evaluation of the "Heatmap + MCTS" paradigm in solving the TSP. We aim to assess the distinct contributions of heatmap quality and MCTS configuration to solver performance, ensuring fair and robust comparisons.

### 3.1 EVALUATION OBJECTIVES

Our evaluation is structured to answer three central questions:

**Q1:** To what extent does the configuration of the MCTS component influence solution quality when applied to diverse heatmap generation techniques?

**Q2:** Can a simple, parameter-free heatmap with an optimally tuned MCTS match or surpass complex learned heatmaps using default MCTS settings?

**Q3:** Which MCTS hyperparameters are most influential, and how does their impact vary with heatmap type and problem scale?

Addressing these questions requires a methodology that isolates MCTS effects for accurate performance attribution.

### 3.2 ENSURING FAIR COMPARISONS

Comparing "Heatmap + MCTS" TSP solvers is challenging as MCTS performance can be a confounding factor. Fixed MCTS settings in prior work may obscure true heatmap efficacy, as MCTS parameters significantly impact solution quality. A sophisticated heatmap might underperform with poorly tuned MCTS, while a simpler one could excel with optimized search.

To ensure fairness, our methodology mandates dedicated MCTS hyperparameter tuning for *each* evaluated heatmap. This optimizes the search strategy for each heatmap's characteristics, enabling a more accurate assessment of its intrinsic value.

### 3.3 MCTS HYPERPARAMETER TUNING PIPELINE

We employ a streamlined MCTS hyperparameter tuning pipeline for standardized and reproducible evaluations across different heatmap methods and TSP scales.

**Tuning Method.** The pipeline uses grid search over key MCTS hyperparameters. For each heatmap and problem scale (TSP-500, TSP-1000, TSP-10000), configurations are evaluated on a dedicated tuning dataset of synthetic TSP instances. The configuration yielding the best average optimality gap is selected for subsequent test evaluations. This tuning is performed independently for each heatmap.

**Key MCTS Hyperparameters.** Based on prior literature (Fu et al., 2021; Min et al., 2024; Xia et al., 2024) and our own empirical sensitivity analysis, we tune the following key hyperparameters:

- `Alpha`: Exploration coefficient (Equation (3)).
- `Beta`: Edge weight update aggressiveness (Equation (4)).

- `Max_Depth`: Maximum $k$ for $k$-opt moves.
- `Max_Candidate_Num`: Candidate edge set size per node.
- `Param_H`: MCTS simulations per move.
- `Use_Heatmap`: Boolean for using heatmap or not for initial candidate set construction.

The search space for these parameters is detailed in Table 12 in Appendix H.

### 3.4 ANALYTICAL TOOLS FOR HYPERPARAMETER IMPORTANCE

To quantify each MCTS hyperparameter's influence on solution quality, we use **SH**apley **A**dditive exParlanations (SHAP) (Lundberg and Lee, 2017; Lundberg et al., 2020). SHAP values, derived from game theory, attribute performance contributions to each parameter, providing model-agnostic insights into their importance.

### 3.5 EXPERIMENTAL SETUP

**Heatmap Methods Evaluated.**    Our framework is applied to diverse heatmap techniques:

- Learning-based: Att-GCN (Fu et al., 2021), DIMES (Qiu et al., 2022), DIFUSCO (Sun and Yang, 2023), UTSP (Min et al., 2024), Fast-T2T(Li et al., 2024).
- Distance-based parameterized: SoftDist (Xia et al., 2024).
- Baselines: A non-informative Zero heatmap and our proposed GT-Prior (Section 5.1).

Pretrained models or generation code are sourced from original authors where possible.

**Datasets.**    Experiments use synthetic TSP instances (sizes 500, 1000, 10000) with a distinct tuning set for each size (128 instances for TSP-500/1000, 16 for TSP-10000; cities sampled uniformly from $[0, 1]^2$) and test sets sourced from Fu et al. (2021). Generalization is assessed on varied distributions generated following Fang et al. (2024) and TSPLIB (Reinelt, 1991) benchmarks. Ground-truth solutions were obtained from Concorde (Applegate et al., 2009) (TSP-500/1000) or LKH-3 (Helsgaun, 2017) (TSP-10000).

**Evaluation Metrics.**    **1**) *Optimality Gap (Gap)*: Relative solution quality to best-known tours, as in Equation (2); **2**) *Improvement*: Gap reduction post-tuning versus default MCTS settings; **3**) *Time*: Heatmap generation + MCTS execution (with MCTS time controlled by `Time_Limit`).

**Computational Environment.**    All experiments were run on an AMD EPYC 9754 128-Core CPU with 256 GB of memory. MCTS runtime per instance is `Time_Limit` $\times N$ seconds.

This methodology underpins the analyses and conclusions presented subsequently.

## 4    COMPONENT IMPACT I: THE CRITICAL ROLE OF MCTS CONFIGURATION

This section presents the empirical analysis of MCTS hyperparameter impact, leveraging the evaluation framework and tuning pipeline detailed in Section 3.3. We first examine the sensitivity and importance of individual MCTS hyperparameters and then quantify the performance gains achieved through their systematic tuning.

### 4.1    MCTS HYPERPARAMETER SENSITIVITY AND IMPORTANCE

The MCTS hyperparameter tuning pipeline was executed using the search space specified in Table 12 in Appendix H. This space includes configurations inspired by prior works (Fu et al., 2021; Min et al., 2024; Xia et al., 2024) and algorithmic analysis, with default settings highlighted in bold. The impact of these MCTS configurations on TSP solution quality was subsequently analyzed using SHAP values, which attribute performance changes to individual hyperparameters. Positive SHAP values suggest an increase in solution length (worse performance), while negative values indicate a reduction (better performance).

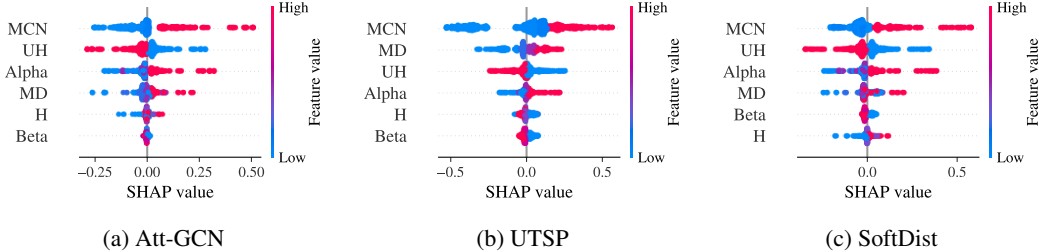

|                |                |                |
| :------------: | :------------: | :------------: |
| (a) Att-GCN    | (b) UTSP       | (c) SoftDist   |

Figure 1: Beeswarm plots of SHAP values for three different heatmaps. MD: `Max_Depth`, MCN: `Max_Candidate_Num`, H: `Param_H`, UH: `Use_Heatmap`. Each dot represents a feature's SHAP value for one instance, indicating its impact on the TSP solution length. The x-axis shows SHAP value magnitude and direction, while the y-axis lists features. Vertical stacking indicates similar impacts across instances. Wider spreads suggest greater influence and potential nonlinear effects. Dot color represents the corresponding feature value.

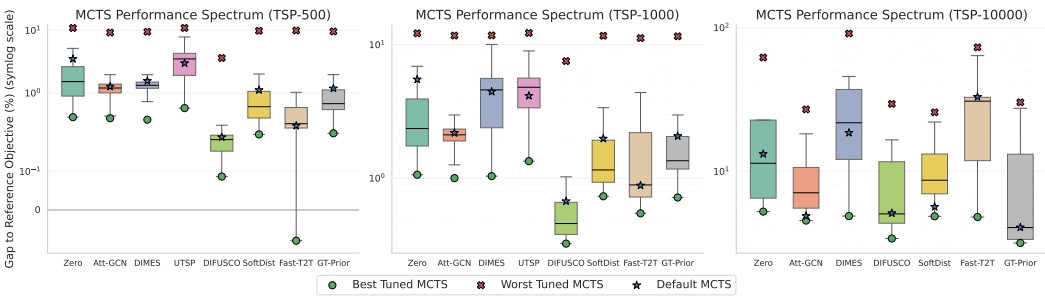

Figure 2: Box plots of the optimality gap (%) for various heatmap sources, scales, and MCTS settings.

Figure 1 presents the SHAP value distributions for key MCTS hyperparameters across three representative heatmap models (Att-GCN, UTSP, and SoftDist) on TSP-500 instances. Additional plots for other models and problem sizes are provided in Appendix E. The `Max_Candidate_Num` parameter consistently demonstrates a strong, often positive, impact across these models, suggesting that reducing the candidate set size from very large defaults can improve both computational speed and solution quality. `Max_Depth` generally exhibits positive SHAP values, indicating that excessively deep $k$-opt explorations within MCTS can sometimes be detrimental to finding good solutions quickly. The parameters `Alpha` (exploration coefficient) and `Use_Heatmap` (determining initial candidate set construction) show mixed effects, revealing non-linear interactions where their optimal values and impact depend on the specific heatmap being used. For instance, `Beta` (edge weight update aggressiveness) shows a notable positive influence in the SoftDist model, implying that its default update strategy might be suboptimal. Conversely, `Param_H` (MCTS simulations per move) generally demonstrates minimal overall influence across the examined heatmaps within the tested ranges. These findings directly address **Q3**, pinpointing influential MCTS hyperparameters and the context-dependent nature of their effects.

## 4.2 QUANTIFYING PERFORMANCE GAINS FROM MCTS TUNING

To quantify the impact of MCTS configuration, we tuned hyperparameter sets within the given search space and report their performance spectrum. During hyperparameter tuning, the `Time_Limit` for MCTS was set to 0.1 for TSP-500 and TSP-1000, and 0.01 for TSP-10000. Performance is reported as the *Optimality Gap (Gap)*. UTSP is not evaluated on TSP-10000 due to unavailability of corresponding heatmaps. The Zero heatmap's tuning involved setting `Use_Heatmap` to `False`[1].

Figure 2 illustrates the critical role of MCTS hyperparameter tuning, directly answering **Q1** by demonstrating the extent to which MCTS configuration determines final solution quality across diverse heatmap generation techniques. The plotted gap is computed relative to borrowed test set reference objectives (not strict per-instance optima), so small negative gaps may occasionally appear

---

[1]For the Zero heatmap, `Use_Heatmap` was set to `False`, as it provides no instance-specific information.

(e.g., Fast-T2T on TSP-500). This impact is evident in the vast performance gap between best-tuned (green circles) and worst-tuned (red 'x') configurations; for instance, DIMES on TSP-10000 ranges from a 4.86% gap to a crippling 91.31% based on MCTS settings alone. Consequently, default MCTS configurations (blue stars) are often far from best-performing settings.

In essence, Figure 2 reveals that MCTS configuration is a dominant performance factor. Effective tuning is not only beneficial but crucial, capable of substantially elevating solution quality for all types of heatmaps and enabling even basic priors to achieve strong results. This underscores the necessity of our streamlined MCTS tuning pipeline (Section 3.3) for rigorous evaluations and realizing strong solver performance. The specific MCTS configurations yielding the best-tuned results are in Appendix H.

This one-time hyperparameter tuning (conducted via grid search) is a pre-computation step comparable in effort to training many learning-based heatmap methods and does not affect the MCTS inference time. Further tuning efficiencies can be realized through parallelization or advanced hyperparameter optimization algorithms like SMAC3 (Lindauer et al., 2022), as discussed in Appendix I.

# 5 COMPONENT IMPACT II: RE-EVALUATING HEATMAP SOPHISTICATION WITH A RIGOROUS BASELINE

While Section 4 established the critical role of MCTS configuration, this section evaluates the prevailing view that increasingly sophisticated heatmap models are the primary drivers of performance in the "Heatmap + MCTS" TSP paradigm. We introduce and evaluate a simple, parameter-free baseline, GT-Prior, derived from the intrinsic $k$-nearest neighbor structure of TSP solutions. By comparing GT-Prior (with optimized MCTS) against complex learned heatmaps, we assess whether the pursuit of heatmap complexity consistently yields justifiable performance gains, especially when the search component is already operating effectively, providing an answer to **Q2**. This analysis aims to provide a clearer perspective on the added value of intricate heatmap models and advocate for the inclusion of strong, simple baselines in future methodological comparisons.

## 5.1 THE $k$-NEAREST PRIOR IN TSP

The $k$-nearest prior in TSP posits that optimal tour edges predominantly connect a city to one of its closest neighbors. This empirical observation has been implicitly used in constructing sparse graph inputs for learning models (Fu et al., 2021; Sun and Yang, 2023; Min et al., 2024), yet its direct use as a primary heatmap source has been less explored.

To elucidate the $k$-nearest prior, we conducted a comprehensive analysis of (near-) optimal solutions for TSP instances of various sizes. Given a set of TSP instances $\mathcal{I}$, for each instance $I \in \mathcal{I}$ and its optimal solution, we calculate the rank of the nearest neighbors for the next city: $k \in \{1, 2, ..., N\}$, and count their occurrences $n_k^I$, where $n_k^I$ represents the number of times of selecting the $k$-nearest cities in an instance's optimal solution. We then calculate the distribution:

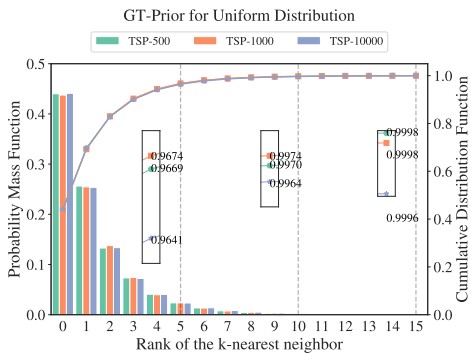

Figure 3: Empirical distribution of $k$-nearest neighbor selection in optimal TSP tours

$$\mathbb{P}_N^I(k) = \frac{n_k^I}{N}, k \in \{1, 2, ..., N\} \tag{5}$$

and average these distributions across all instances to derive the empirical distribution:

$$\hat{\mathbb{P}}_N(k) = \frac{1}{|\mathcal{I}|} \sum_{I \in \mathcal{I}} \mathbb{P}_N^I(k), k \in \{1, 2, ..., N\}. \tag{6}$$

To quantify this prior empirically, we analyzed (near-)optimal solutions for uniform TSP instances of varying sizes (TSP-500, TSP-1000 using Concorde; TSP-10000 using LKH-3). For each instance $I$

Table 1: Results on large-scale TSP problems. Abbreviations: RL (Reinforcement learning), SL (Supervised learning), UL (Unsupervised learning), AS (Active search), G (Greedy decoding), S (Sampling decoding), BS (Beam-search). * indicates baseline for performance gap. † indicates methods using heatmaps of test set from Xia et al. (2024) with our MCTS setup. Some methods show two *Time* terms (heatmap generation and MCTS runtimes). MCTS times denote the equivalent sequential runtime per instance. Concorde and Gurobi results are sourced from Fu et al. (2021); Qiu et al. (2022).

| METHOD | TYPE | TSP-500 | | | TSP-1000 | | | TSP-10000 | | |
|---|---|---|---|---|---|---|---|---|---|---|
| | | LENGTH ↓ | GAP ↓ | TIME ↓ | LENGTH ↓ | GAP ↓ | TIME ↓ | LENGTH ↓ | GAP ↓ | TIME ↓ |
| CONCORDE | OR(EXACT) | 16.55* | — | 17.65S | 23.12* | — | 3.12M | N/A | N/A | N/A |
| GUROBI | OR(EXACT) | 16.55 | 0.00% | 21.39M | N/A | N/A | N/A | N/A | N/A | N/A |
| LKH-3 (DEFAULT) | OR(HEURISTIC) | 16.55 | 0.00% | 14.84S | 23.12 | 0.00% | 1.02M | 71.77* | — | 28.73M |
| ZERO | MCTS | 16.66 | 0.66% | 0.00M+ 50.06S | 23.39 | 1.16% | 0.00M+ 1.67M | 74.50 | 3.80% | 0.00M+ 16.65M |
| ATT-GCN† | SL+MCTS | 16.66 | 0.69% | 0.52M+ 50.06S | 23.37 | 1.09% | 0.73M+ 1.67M | 73.95 | 3.03% | 4.16M+ 16.65M |
| DIMES† | RL+MCTS | 16.66 | 0.43% | 0.97M+ 50.06S | 23.37 | 1.11% | 2.08M+ 1.67M | 73.97 | 3.06% | 4.65M+ 16.65M |
| UTSP† | UL+MCTS | 16.69 | 0.90% | 1.37M+ 50.06S | 23.47 | 1.53% | 3.35M+ 1.67M | — | — | — |
| SOFTDIST† | SOFTDIST+MCTS | 16.62 | 0.43% | 0.00M+ 50.06S | 23.30 | 0.80% | 0.00M+ 1.67M | 73.89 | 2.95% | 0.00M+ 16.65M |
| DIFUSCO† | SL+MCTS | 16.60 | 0.33% | 3.61M+ 50.06S | **23.24** | **0.53%** | 11.86M+ 1.67M | 73.47 | 2.37% | 28.51M+ 16.65M |
| FAST-T2T | SL+MCTS | **16.57** | **0.12%** | 0.50M+ 50.06S | 23.27 | 0.65% | 1.78M+ 1.67M | 74.80 | 4.22% | 7.73M+ 16.65M |
| GT-PRIOR | PRIOR+MCTS | 16.63 | 0.50% | 0.00M+ 50.06S | 23.31 | 0.85% | 0.00M+ 1.67M | **73.31** | **2.14%** | 0.00M+ 16.65M |

from a set $\mathcal{I}$, and its (near-)optimal tour, we computed the frequency $n_k^I$ with which an edge connects to the $k$-th nearest neighbor. The averaged empirical distribution $\hat{\mathbb{P}}_N(k) = \frac{1}{|\mathcal{I}|} \sum_{I \in \mathcal{I}} \frac{n_k^I}{N}$ is shown in Figure 3. The results reveal a strong locality: the probability of selecting one of the top 5 nearest neighbors exceeds 94%, rising above 99% for the top 10. This distribution is highly consistent across TSP scales, a finding that also holds for instances from different underlying distributions (see Appendix J for similar results).

Leveraging insights from the optimal solution, we construct the heatmap by assigning probabilities to edges based on the empirical distribution of the $k$-nearest prior $\hat{\mathbb{P}}_N(\cdot)$. For each city $i$ in a TSP instance of size $N$, we assign probabilities to edges $(i, j)$ as follows:

$$P_{ij}^N = \hat{\mathbb{P}}_N(k_{ij}), k_{ij} \in \{1, 2, ..., N\} \tag{7}$$

where $k_{ij}$ is the rank of city $j$ among $i$'s neighbors in terms of proximity (see the detailed statistical results in Appendix L). Importantly, this heatmap is parameter-free and scale-independent, thus requiring no tuning or learning phase.

## 5.2 PERFORMANCE DEMONSTRATION: CHALLENGING COMPLEXITY

We evaluated GT-Prior against various heatmap methods, all coupled with MCTS configurations tuned according to our pipeline (Section 3.3). This ensures that comparisons reflect the heatmap's intrinsic quality when its search partner is optimized, rather than differences in MCTS efficacy.

As shown in Table 1, GT-Prior, a simple parameter-free heatmap, when combined with an optimally tuned MCTS, achieves performance highly competitive with, and in some cases (TSP-10000) superior to, far more complex learning-based heatmap generators like DIFUSCO. For TSP-500, TSP-1000, and TSP-10000, GT-Prior yields optimality gaps of 0.50%, 0.85%, and 2.14%, respectively. This performance is achieved with no heatmap generation time at inference, similar to SoftDist and Zero.

Critically, the Zero heatmap, providing no edge guidance, still achieves respectable gaps (e.g., 0.66% for TSP-500) solely through tuned MCTS (where `Use_Heatmap` is optimally set to `False`, relying on distance for candidate selection). This underscores the substantial impact of the search component itself. The strong showing of GT-Prior and even the tuned Zero heatmap challenges the narrative that gains in TSP solutions primarily hinge on increasing heatmap model sophistication. It suggests that much of the solution quality can be attributed to a well-calibrated search process acting on

Table 2: Generalization performance of different methods trained on TSP-500 across varying TSP sizes (TSP-500, TSP-1000, TSP-10000). "Res Type" refers to the result type: "Ori." indicates the performance on the same scales during the test phase, while "Gen." represents the model's generalized performance on different scales.

| METHOD | RES TYPE | TSP-500 | | TSP-1000 | | TSP-10000 | |
|--------|----------|---------|------------|----------|------------|-----------|------------|
| | | GAP ↓ | DEGRADATION ↓ | GAP ↓ | DEGRADATION ↓ | GAP ↓ | DEGRADATION ↓ |
| DIMES | ORI./GEN. | 0.43%/0.43% | 0.00% | 1.11%/1.19% | 0.08% | 3.05%/4.29% | 1.24% |
| UTSP | ORI./GEN. | 0.90%/0.90% | 0.00% | 1.53%/1.44% | -0.09% | — | — |
| DIFUSCO | ORI./GEN. | 0.33%/0.33% | 0.00% | 0.53%/0.86% | 0.33% | 2.36%/5.27% | 2.91% |
| SOFTDIST | ORI./GEN. | 0.43%/0.43% | 0.00% | 0.80%/0.97% | 0.17% | 2.94%/3.90% | 0.96% |
| FAST-T2T | ORI./GEN. | 0.12%/0.12% | 0.00% | 0.65%/1.40% | 0.75% | 4.22%/4.16% | -0.06% |
| GT-PRIOR | ORI./GEN. | 0.50%/0.50% | 0.00% | 0.85%/0.88% | 0.03% | 2.14%/2.13% | -0.01% |

Table 3: Optimality gap (%, ↓) across distributions and sizes. Lighter color indicates lower gap.

| METHOD | TSP-500 | | | TSP-1000 | | | TSP-10000 | | |
|--------|---------|-----------|-----------|----------|-----------|-----------|-----------|-----------|-----------|
| | CLUSTER | EXPLOSION | IMPLOSION | CLUSTER | EXPLOSION | IMPLOSION | CLUSTER | EXPLOSION | IMPLOSION |
| ZERO | 0.79 | 0.58 | 0.67 | 1.22 | 1.14 | 1.23 | 1.81 | 2.53 | 2.57 |
| ATT-GCN | 0.74 | 0.58 | 0.65 | 1.10 | 0.96 | 1.00 | 1.12 | 1.57 | 1.59 |
| DIMES | 0.90 | 0.62 | 0.72 | 1.28 | 1.06 | 1.16 | 2.23 | 2.90 | 2.91 |
| UTSP | 0.97 | 0.72 | 0.83 | 1.38 | 1.25 | 1.36 | | — | |
| DIFUSCO | 0.88 | 0.65 | 0.74 | 0.53 | 0.42 | 0.32 | 1.96 | 2.50 | 2.42 |
| SOFTDIST | 0.98 | 0.56 | 0.49 | 1.55 | 1.03 | 0.75 | 1.45 | 1.72 | 0.33 |
| FAST-T2T | 0.28 | 0.17 | 0.20 | 0.57 | 0.94 | 0.69 | 2.13 | 3.24 | 3.22 |
| GT-PRIOR | 0.51 | 0.38 | 0.49 | 0.70 | 0.63 | 0.74 | 0.35 | 0.93 | 0.58 |

fundamental problem characteristics, a point potentially understated in evaluations that do not tune MCTS for simpler baselines.

## 5.3 GENERALIZATION ABILITY: ROBUSTNESS OF SIMPLICITY

We further assessed the generalization of GT-Prior by applying the prior derived from TSP-500 data to larger TSP instances (TSP-1000, TSP-10000), as well as different distributions (derived from uniform and tested on other distributions), comparing against learned models under the same cross-scale evaluation.

Table 2 reveals GT-Prior's robust generalization across scales. When the prior derived from uniform TSP-500 is applied to TSP-10000, GT-Prior's performance degradation is minimal (merely a -0.01% change in gap, effectively maintaining its 2.13% gap), substantially outperforming complex learned models like DIFUSCO, which sees its gap increase from 2.36% to 5.27% (a 2.91% degradation). This suggests that simpler priors based on inherent problem structure (like $k$-nearest neighbors) may offer greater robustness and scalability than intricate learned patterns, which might overfit to training distributions or scales.

This robustness extends to generalization across qualitatively different problem structures. As evidenced in Table 3, when MCTS settings (tuned on uniform data, and the heatmap also derived from uniform data) are applied to instances from clustered, explosion, and implosion distributions, GT-Prior consistently maintains strong performance. For example, on TSP-10000, GT-Prior achieves impressive optimality gaps of 0.35% (clustered), 0.93% (explosion), and 0.58% (implosion). These results frequently surpass those of more complex models like DIFUSCO (1.96%, 2.50%, 2.42% respectively) under these challenging cross-distribution test conditions. This highlights that GT-Prior's fundamental $k$-nearest neighbor prior is less susceptible to distributional shifts than learned patterns, which might inadvertently specialize to the characteristics of (typically uniform) training data. While sophisticated learning-based models can achieve excellent results in certain cases, demonstrating the generalization ability of their learned features (e.g., DIFUSCO's strong performance on TSP-1000 across distributions), GT-Prior's consistent efficacy underscores the value of simple, structurally-grounded priors for achieving reliable generalization—a key quality for practical and versatile TSP solvers. This resilience is a crucial evaluative aspect, particularly for solvers intended for diverse, large-scale applications.

More generalization results of models trained on TSP-1000 and TSP-10000 are left in Appendix F, and additional results on TSPLIB instances are listed in Appendix G.

## 6 CONCLUSIONS

This study underscores the necessity of a more balanced and rigorous approach to the "Heatmap + MCTS" paradigm for the TSP. By empirically demonstrating the distinct impact of MCTS configurations and the competitive strength of a simple, parameter-free $k$-nearest prior when coupled with optimized search, our work challenges the prevailing emphasis on heatmap sophistication. The introduced streamlined MCTS hyperparameter tuning pipeline offers a concrete pathway toward fairer and more insightful comparisons of future heatmap designs. Looking ahead, these evaluative insights encourage a research trajectory that moves beyond isolated component optimization. By fostering a synergistic, holistically understood, and optimized integration of learning and search, the field can develop TSP solvers that are not only high-performing but also more robust, efficient, and genuinely impactful.

### ETHICS STATEMENT

The research presented in this paper, aimed at improving solutions for the Traveling Salesman Problem (TSP), has several potential societal impacts. On the positive side, advancements in TSP solvers can directly enhance efficiency in sectors like logistics and manufacturing, leading to reduced fuel consumption, lower operational costs, and decreased environmental emissions. Our findings that simpler, well-tuned methods can be highly effective may also help democratize access to advanced optimization tools, allowing smaller entities to benefit without requiring massive computational resources.

Conversely, we acknowledge potential negative impacts. As with many advancements in AI and automation, the increased efficiency from improved TSP solvers could contribute to job displacement in manual planning and routing roles. There is also a risk of unintended consequences or exacerbating existing inequities if these tools are deployed without careful consideration of all relevant factors. Therefore, this work encourages a methodical approach to building AI systems, emphasizing the importance of understanding and tuning all components. By demonstrating the power of simpler priors combined with careful search configuration, we advocate for solutions that are more transparent and robust, aligning with principles of responsible AI development. Continuous attention to fairness, robustness, and human oversight will be crucial as such technologies are deployed.

### REPRODUCIBILITY STATEMENT

To ensure the reproducibility of our research, we provide a comprehensive suite of resources. The complete source code, including scripts for hyperparameter tuning and result evaluation, is available at: `https://github.com/LOGO-CUHKSZ/beyond-heatmap-mcts-tsp`. A detailed README file offers step-by-step instructions for setting up the computational environment and executing the experiments described in this paper. Our full evaluation methodology, experimental setup, datasets, and computational environment are described in Section 3.5. The specific MCTS hyperparameter search space and the best-tuned configurations used to produce our final results are detailed in the Appendix H and I. The procedure for generating our proposed GT-Prior is explained in Section 5.1. The datasets used in our experiments were sourced from established benchmarks as cited in Section 3.5, and the generation code for synthetic instances is also provided.

### ACKNOWLEDGMENTS

This work was supported by National Science and Technology Major Project under Grant 2022ZD0116408.

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

## A  ADDITIONAL RELATED WORKS

Approaches using machine learning to address the Traveling Salesman Problem (TSP) generally fall into two distinct groups based on how they generate solutions. The first group, known as construction methods, incrementally forms a path by sequentially adding cities to an unfinished route, following an autoregressive process until the entire path is completed. The second group, improvement methods, starts with a complete route and continually applies local search operations to improve the solution.

**Construction Methods**  Since Vinyals et al. (2015); Bello et al. (2016) introduced the autoregressive combinatorial optimization neural solver, numerous advancements have emerged in subsequent years (Deudon et al., 2018; Kool et al., 2019; Peng et al., 2020; Kwon et al., 2020; 2021). These include enhanced network architectures (Kool et al., 2019), more sophisticated deep reinforcement learning techniques (Khalil et al., 2017; Ma et al., 2019; Choo et al., 2022), and improved training methods (Kim et al., 2022; Bi et al., 2022). For large-scale TSP, Pan et al. (2023) adopts a hierarchical divide-and-conquer approach, breaking down the complex TSP into more manageable open-loop TSP sub-problems.

**Improvement Methods**  In contrast to construction methods, improvement-based solvers leverage neural networks to progressively refine an existing feasible solution, continuing the process until the computational limit is reached. These improvement methods are often influenced by traditional local search techniques like $k$-opt, and have been shown to deliver impressive results in various previous studies (Chen and Tian, 2019; Wu et al., 2021; Kim et al., 2021; Hudson et al., 2021). Ye et al. (2024) implements a divide-and-conquer approach, using search-based methods to enhance the solutions of smaller subproblems generated from the larger instances.

Recent breakthroughs in solving large-scale TSP problems (Fu et al., 2021; Qiu et al., 2022; Sun and Yang, 2023; Min et al., 2024; Xia et al., 2024) have incorporated Monte Carlo tree search (MCTS) as an effective post-processing technique. These heatmaps serve as priors for guiding the MCTS, resulting in impressive performance in large-scale TSP solutions, achieving state-of-the-art results.

**Other Directions**  In addition to exploring solution methods for combinatorial optimization problems, some studies investigate intrinsic challenges encountered during the learning phase. These include generalization issues during inference (Wang et al., 2021; Zhou et al., 2023; Wang et al., 2024) and multi-task learning (Wang and Yu, 2023; Liu et al., 2024; Zhou et al., 2024) aimed at developing foundational models.

## B  IMPACT OF HEURISTIC POSTPROCESSING

In our experimental reproduction of various learning-based heatmap generation methods for the Traveling Salesman Problem (TSP), we identified a critical yet often overlooked factor affecting performance: the post-processing of model-generated heatmaps. This section details the post-processing strategies employed by different methods and evaluates their impact on performance metrics.

### B.1  POSTPROCESSING STRATEGIES

**DIMES**  DIMES generates an initial heatmap matrix of dimension $n \times n$ from a $k$-nearest neighbors ($k$-NN) subgraph of the original TSP instance ($k = 50$). The post-processing involves two steps:

1. Sparsification: Retaining only the top-5 values for each row, setting all others to a significantly negative number.

2. Adaptive $\mathrm{softmax}$: Iteratively applying a temperature-scaled $\mathrm{softmax}$ function with gradual temperature reduction until the minimum non-zero probability exceeds a predefined threshold.

**DIFUSCO**  DIFUSCO also generates a sparse heatmap based on the $k$-NN subgraph ($k = 50$ for TSP-500, $k = 100$ for larger scales). The post-processing differs based on problem scale:

Table 4: Performance Degradation for Different Methods with and without Postprocessing on TSP-500, TSP-1000, and TSP-10000. 'W' indicates with postprocessing, while 'W/O' indicates without postprocessing.

| METHOD | POSTPROCESSING | TSP-500 | | TSP-1000 | | TSP-10000 | |
|---|---|---|---|---|---|---|---|
| | | GAP $\downarrow$ | DEGRADATION $\downarrow$ | GAP $\downarrow$ | DEGRADATION $\downarrow$ | GAP $\downarrow$ | DEGRADATION $\downarrow$ |
| DIMES | W/O | 2.50% | 0.93% | 9.07% | 6.77% | 15.87% | 12.81% |
| | W | 1.57% | | 2.30% | | 3.05% | |
| UTSP | W/O | 4.50% | 1.36% | 6.30% | 2.10% | — | — |
| | W | 3.14% | | 4.20% | | | |
| DIFUSCO | W/O | 2.33% | 1.88% | 0.66% | -0.40% | 45.20% | 42.52% |
| | W | 0.45% | | 1.07% | | 2.69% | |

1. For TSP-500 and TSP-1000: A single step integrating Euclidean distances, thresholding, and symmetrization.

2. For TSP-10000: Two steps are applied sequentially: a) Additional supervision using a greedy decoding strategy followed by 2-opt heuristics. b) The same process as used for smaller instances.

**UTSP** UTSP's post-processing is straightforward, involving sparsification of the dense heatmap matrix by preserving only the top 20 values per row.

**Fast-T2T** We use minimal post-processing: apply $\mathrm{softmax}$ to final-step logits and take the edge-existence probability as the heatmap. For sparse $k$-NN outputs, we map scores to a dense $n \times n$ matrix and symmetrize by $\max(H_{ij}, H_{ji})$ before MCTS.

*Note.* The original Fast-T2T (Li et al., 2024) paper does not provide MCTS-specific results. Therefore, we adopt the simple post-processing above for our reproduction.

### B.2 EXPERIMENTAL RESULTS

We conducted experiments on the test set for heatmaps generated by these three methods, both with and without post-processing, using the default MCTS setting. Results are presented in Table 4.

Our findings reveal that heatmaps generated without post-processing generally exhibit performance degradation, particularly for TSP-10000, where the gap increases by orders of magnitude. This underscores the importance of sparsification for large-scale instances and highlights the tendency of existing methodologies to overstate their efficacy in training complex deep learning models.

Interestingly, DIFUSCO's heatmap without post-processing outperforms its post-processed counterpart for TSP-1000, suggesting that the DIFUSCO model, when well-trained on this scale, can generate helpful heatmap matrices to guide MCTS without additional refinement.

These results emphasize the critical role of post-processing in enhancing the performance of learning-based heatmap generation methods for TSP, particularly as problem scales increase. They also highlight the need for careful evaluation of model outputs and the potential for over-reliance on post-processing to mask limitations in model training and generalization.

The substantial performance gap between heatmaps with and without post-processing raises questions about the extent to which the reported performance gains can be attributed solely to the learning modules of these methods. While the learning components undoubtedly contribute to the overall effectiveness, the significant impact of post-processing suggests that the raw output of the learning models may not be as refined or directly applicable as previously thought.

In light of these findings, we recommend that future research on heatmap-based methods for TSP provide a detailed description of their post-processing operations. Additionally, we suggest reporting results both with and without post-processing to offer a more comprehensive understanding of the method's performance and the relative contributions of its learning and post-processing components. This approach would foster greater transparency in the field and facilitate more accurate comparisons between different methodologies.

## C    ANALYSIS OF ONE-OFF COMPUTATIONAL COSTS

To ensure a holistic comparison, we analyze the one-off setup costs: model training for learning-based baselines versus hyperparameter tuning for MCTS.

**Training Costs.** Table 5 summarizes the training times reported in original papers. Deep learning methods typically incur heavy computational overheads, requiring significant GPU hours (e.g., $\sim$10 hours for DIMES on TSP-10000).

Table 5: Approximate training times for learning-based methods. Note: Times are rough references due to hardware variances.

| Method | TSP-500 | TSP-1000 | TSP-10000 |
|--------|---------|----------|-----------|
| Att-GCN | $\sim$25 h | $\sim$25 h | $\sim$25 h |
| DIMES | $\sim$1.5 h | $\sim$1.7 h | $\sim$10 h |
| UTSP | $\sim$0.5 h | N/A | N/A |

**Tuning vs. Training.** MCTS tuning is a comparable one-off cost but is generally more resource-efficient. Using SMAC3 (detailed in Appendix I), tuning MCTS for TSP-10000 requires only **3.47 hours on a CPU**. This is notably lower than the GPU-intensive training required for baselines like DIMES. Furthermore, the resulting configurations (Table 13) are reusable across similar distributions.

**GT-Prior Construction.** The cost of constructing GT-Prior is negligible. It requires solving only a small set of instances (e.g., 128 for TSP-500) to extract $k$-NN statistics, avoiding the expensive pre-training phase entirely.

## D    FULL EXPERIMENTAL RESULTS

The following table presents the complete results of the large-scale TSP problems, including the four end-to-end learning-based methods that were previously omitted in the main paper due to space constraints. These methods include EAN (d O Costa et al., 2020), AM (Kool et al., 2019), GCN (Joshi et al., 2019), and POMO+EAS (Hottung et al., 2021). We also included a more recent heatmap method: Fast-T2T(Li et al., 2024). The methods listed here employ reinforcement learning (RL), supervised learning (SL), and unsupervised learning (UL) techniques, in addition to various decoding strategies such as greedy, sampling, and beam-search.

Complementing these, we introduce specific comparisons with greedy decoding and plain 2-opt to isolate the impact of the search mechanism. The greedy results exhibit large optimality gaps (e.g., >50%), confirming that myopic decisions inevitably discard critical distributional information found in the heatmaps. Similarly, while plain 2-opt improves solution quality, it lags significantly behind MCTS on large-scale instances (TSP-10000). This performance gap highlights that local search alone is insufficient to escape local optima at this scale, validating the necessity of MCTS for providing high-level global guidance.

Table 6: Full Results on large-scale TSP problems. Abbreviations: RL (Reinforcement learning), SL (Supervised learning), UL (Unsupervised learning), AS (Active search), G (Greedy decoding), S (Sampling decoding), and BS (Beam-search). ∗ indicates the baseline for performance gap calculation. † indicates methods utilizing heatmaps provided by Xia et al. (2024), with MCTS executed on our setup. Some methods list two terms for *Time*, corresponding to heatmap generation and MCTS runtimes, respectively. Concorde and Gurobi results are sourced from Fu et al. (2021); Qiu et al. (2022).

| Method | Type | TSP-500 | | | TSP-1000 | | | TSP-10000 | | |
|---|---|---|---|---|---|---|---|---|---|---|
| | | Length ↓ | Gap ↓ | Time ↓ | Length ↓ | Gap ↓ | Time ↓ | Length ↓ | Gap ↓ | Time ↓ |
| Concorde | OR(exact) | 16.55* | — | 17.65s | 23.12* | — | 3.12m | N/A | N/A | N/A |
| Gurobi | OR(exact) | 16.55 | 0.00% | 21.39m | N/A | N/A | N/A | N/A | N/A | N/A |
| LKH-3 (default) | OR(heuristic) | 16.55 | 0.00% | 14.84s | 23.12 | 0.00% | 1.02m | 71.77* | — | 28.73m |
| Nearest Insertion | OR | 20.62 | 24.59% | 0.00s | 28.96 | 25.26% | 0.00s | 90.51 | 26.12% | 0.38s |
| Random Insertion | OR | 18.57 | 12.21% | 0.00s | 26.12 | 12.98% | 0.00s | 81.85 | 14.05% | 0.25s |
| Farthest Insertion | OR | 18.30 | 10.57% | 0.00s | 25.72 | 11.25% | 0.00s | 80.59 | 12.30% | 0.38s |
| EAN | RL+S | 28.63 | 73.03% | 9.46s | 50.30 | 117.59% | 17.38s | N/A | N/A | N/A |
| EAN | RL+S+2-opt | 23.75 | 43.57% | 27.07s | 47.73 | 106.46% | 2.53m | N/A | N/A | N/A |
| AM | RL+S | 22.64 | 36.84% | 7.33s | 42.80 | 85.15% | 29.99s | 431.58 | 501.31% | 47.36s |
| AM | RL+G | 20.02 | 20.99% | 0.71s | 31.15 | 34.75% | 1.49s | 141.68 | 97.40% | 22.46s |
| AM | RL+BS | 19.53 | 18.03% | 10.31s | 29.90 | 29.23% | 46.12s | 129.40 | 80.29% | 6.79m |
| GCN | SL+G | 29.72 | 79.61% | 3.13s | 48.62 | 110.29% | 13.37s | N/A | N/A | N/A |
| GCN | SL+BS | 30.37 | 83.55% | 17.82s | 51.26 | 121.73% | 24.22s | N/A | N/A | N/A |
| POMO+EAS-Emb | RL+AS | 19.24 | 16.25% | 6.00m | N/A | N/A | N/A | N/A | N/A | N/A |
| POMO+EAS-Lay | RL+AS | 19.35 | 16.92% | 7.59m | N/A | N/A | N/A | N/A | N/A | N/A |
| POMO+EAS-Tab | RL+AS | 24.54 | 48.22% | 5.44m | 49.56 | 114.36% | 29.74m | N/A | N/A | N/A |
| Zero | MCTS | 16.66 | 0.66% | 0.00m+ 50.06s | 23.39 | 1.16% | 0.00m+ 1.67m | 74.50 | 3.80% | 0.00m+ 16.65m |
| Att-GCN† | SL+MCTS | 16.66 | 0.69% | 0.52m+ 50.06s | 23.37 | 1.09% | 0.73m+ 1.67m | 73.95 | 3.03% | 4.16m+ 16.65m |
| | SL+2-opt | 17.42 | 5.27% | 0.52m+ 50.06s | 24.77 | 7.16% | 0.73m+ 1.67m | 79.48 | 10.73% | 4.16m+ 16.65m |
| | SL+Greedy | 30.71 | 85.63% | 0.52m+ 0.02s | 50.65 | 119.11% | 0.73m+ 0.05s | 304.88 | 324.76% | 4.16m+ 2.13s |
| DIMES† | RL+MCTS | 16.66 | 0.43% | 0.97m+ 50.06s | 23.37 | 1.11% | 2.08m+ 1.67m | 73.97 | 3.06% | 4.65m+ 16.65m |
| | RL+2-opt | 17.58 | 6.26% | 0.97m+ 50.06s | 25.00 | 8.15% | 2.08m+ 1.67m | 93.56 | 30.35% | 4.65m+ 16.65m |
| | RL+Greedy | 51.43 | 210.82% | 0.97m+ 0.02s | 95.18 | 311.72% | 2.08m+ 0.05s | 771.78 | 975.24% | 4.65m+ 2.17s |
| UTSP† | UL+MCTS | 16.69 | 0.90% | 1.37m+ 50.06s | 23.47 | 1.53% | 3.35m+ 1.67m | — | — | — |
| | UL+2-opt | 17.59 | 6.32% | 1.37m+ 50.06s | 25.03 | 8.28% | 3.35m+ 1.67m | — | — | — |
| | UL+Greedy | 25.48 | 54.00% | 1.37m+ 0.02s | 39.46 | 70.70% | 3.35m+ 0.05s | — | — | — |
| SoftDist† | SoftDist+MCTS | 16.62 | 0.43% | 0.00m+ 50.06s | 23.30 | 0.80% | 0.00m+ 1.67m | 73.89 | 2.95% | 0.00m+ 16.65m |
| | SoftDist+2-opt | 17.50 | 5.75% | 0.00m+ 50.06s | 24.82 | 7.34% | 0.00m+ 1.67m | 79.10 | 10.21% | 0.00m+ 16.65m |
| | SoftDist+Greedy | 20.87 | 26.13% | 0.00m+ 0.02s | 29.06 | 25.69% | 0.00m+ 0.05s | 91.43 | 27.39% | 0.00m+ 2.26s |
| DIFUSCO† | SL+MCTS | 16.60 | 0.33% | 3.61m+ 50.06s | **23.24** | **0.53%** | 11.86m+ 1.67m | 73.47 | 2.37% | 28.51m+ 16.65m |
| | SL+2-opt | 16.69 | 0.89% | 3.61m+ 50.06s | 24.38 | 5.45% | 11.86m+ 1.67m | 78.78 | 9.76% | 28.51m+ 16.65m |
| | SL+Greedy | 18.92 | 14.36% | 3.61m+ 0.02s | 36.90 | 59.61% | 11.86m+ 0.05s | 120.53 | 67.92% | 28.51m+ 2.13s |
| Fast-T2T | SL+MCTS | **16.57** | **0.12%** | 0.50m+ 50.06s | 23.27 | 0.65% | 1.78m+ 1.67m | 74.80 | 4.22% | 7.73m+ 16.65m |
| | SL+2-opt | 16.71 | 0.99% | 0.50m+ 50.06s | 23.92 | 3.48% | 1.78m+ 1.67m | 112.99 | 57.42% | 7.73m+ 16.65m |
| | SL+Greedy | 18.32 | 10.75% | 0.50m+ 0.02s | 25.89 | 11.98% | 1.78m+ 0.05s | 101.05 | 40.78% | 7.73m+ 2.23s |
| GT-Prior | Prior+MCTS | 16.63 | 0.50% | 0.00m+ 50.06s | 23.31 | 0.85% | 0.00m+ 1.67m | **73.31** | **2.14%** | 0.00m+ 16.65m |
| | Prior+2-opt | 17.54 | 5.99% | 0.00m+ 50.06s | 24.91 | 7.77% | 0.00m+ 1.67m | 79.39 | 10.61% | 0.00m+ 16.65m |
| | Prior+Greedy | 25.69 | 55.27% | 0.00m+ 0.02s | 40.45 | 74.99% | 0.00m+ 0.05s | 197.77 | 175.53% | 0.00m+ 2.26s |

# E  ADDITIONAL HYPERPARAMETER IMPORTANCE ANALYSIS

We employed the SHAP method to analyze hyperparameter importance across all conducted grid search experiments. Most resulting beeswarm plots for TSP-500, TSP-1000, and TSP-10000 are in Figure 4 (including 'Zero' heatmap where `Use_Heatmap` is set to `False`). Plots of the UTSP heatmap are presented in Figure 5.

The patterns of TSP-1000 are similar to those of TSP-500, as discussed in Section 4.1. However, the patterns for TSP-10000 show a major difference, where the influence of `Max_Candidate_Num` and `Use_Heatmap` becomes dominant. Furthermore, their SHAP values are clearly clustered rather than continuous, as observed in smaller scales. This could be explained by the candidate set of large-scale TSP instances having a major impact on the running time of MCTS $k$-opt search. Additionally, the time limit setting causes the performance of different hyperparameter settings for `Max_Candidate_Num` and `Use_Heatmap` to become more distinct.

# F  ADDITIONAL GENERALIZATION ABILITY RESULTS

Table 7 presents additional results on the generalization ability of various methods when trained on TSP-1000 and TSP-10000, respectively.

For models trained on TSP-1000, GT-Prior continues to demonstrate superior generalization capability. When generalizing to smaller instances (TSP-500), GT-Prior shows minimal performance degradation (0.02%), comparable to DIMES and better than UTSP and SoftDist. For larger instances (TSP-10000), GT-Prior maintains consistent performance with a slight improvement (-0.02% degradation), outperforming all other methods. DIFUSCO, while showing good performance on TSP-500 and TSP-1000, experiences significant degradation (2.91%) when scaling to TSP-10000.

The results for models trained on TSP-10000 further highlight GT-Prior's robust generalization ability. When applied to smaller problem sizes (TSP-500 and TSP-1000), GT-Prior exhibits minimal performance degradation (0.01% and 0.02%, respectively). In contrast, other methods show more substantial degradation, particularly for TSP-1000. Notably, SoftDist experiences severe performance deterioration (73.36%) when generalizing to TSP-1000, while DIFUSCO shows significant degradation for both TSP-500 (0.63%) and TSP-1000 (2.74%).

These results consistently demonstrate GT-Prior's exceptional ability to generalize across various problem scales, maintaining stable performance regardless of whether it is scaling up or down from the training instance size. This stability is particularly evident when compared to the other methods, which often struggle with significant performance degradation when generalizing to different problem sizes.

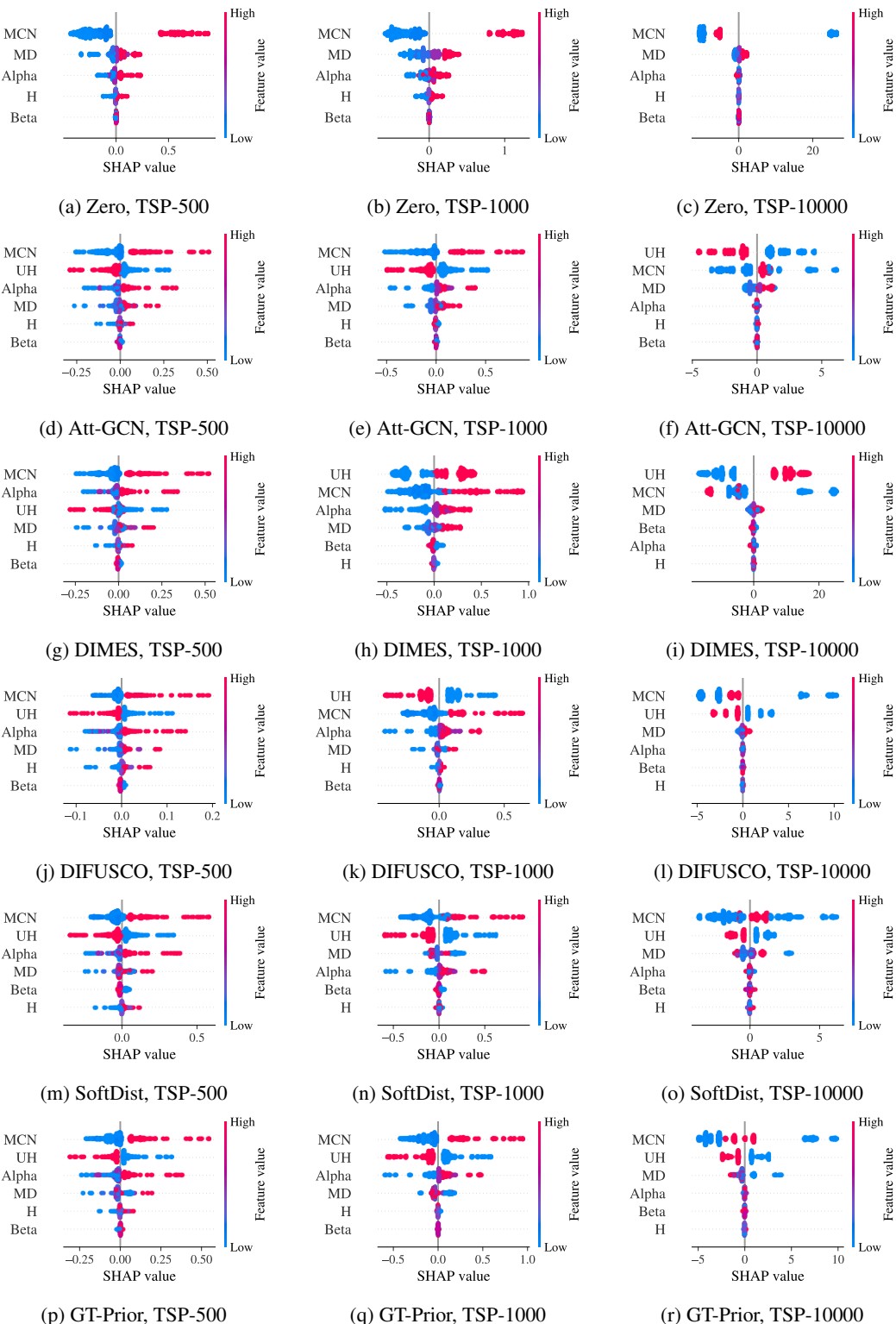

Figure 4: Beeswarm plots of SHAP values for six methods across different TSP sizes.

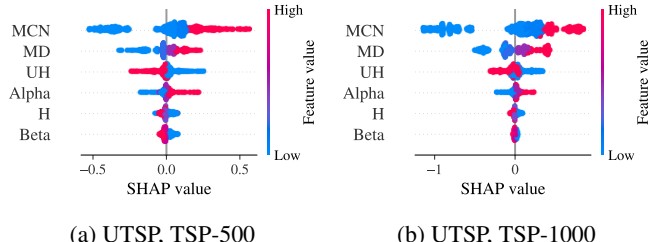

(a) UTSP, TSP-500          (b) UTSP, TSP-1000

Figure 5: Beeswarm plots of SHAP values for the UTSP heatmap across different TSP sizes.

Table 7: Generalization on the model trained on TSP-1000 (the upper table) and TSP-10000 (the lower table).

| METHOD | RES TYPE | TSP-500 | | TSP-1000 | | TSP-10000 | |
|---|---|---|---|---|---|---|---|
| | | GAP↓ | DEGRADATION↓ | GAP↓ | DEGRADATION↓ | GAP↓ | DEGRADATION↓ |
| DIMES | ORI. | 0.69% | 0.02% | 1.11% | 0.00% | 3.05% | 1.01% |
| | GEN. | 0.71% | | 1.11% | | 4.06% | |
| UTSP | ORI. | 0.90% | 0.06% | 1.53% | 0.00% | — | — |
| | GEN. | 0.96% | | 1.53% | | | |
| DIFUSCO | ORI. | 0.33% | -0.07% | 0.53% | 0.00% | 2.36% | 2.91% |
| | GEN. | 0.26% | | 0.53% | | 5.27% | |
| SOFTDIST | ORI. | 0.43% | 0.08% | 0.80% | 0.00% | 2.94% | 0.74% |
| | GEN. | 0.51% | | 0.80% | | 3.68% | |
| FAST-T2T | ORI. | 0.12% | -0.01% | 0.65% | 0.00% | 4.22% | -0.92% |
| | GEN. | 0.11% | | 0.65% | | 3.30% | |
| GT-PRIOR | ORI. | 0.50% | 0.02% | 0.85% | 0.00% | 2.13% | -0.02% |
| | GEN. | 0.52% | | 0.85% | | 2.11% | |

| METHOD | RES TYPE | TSP-500 | | TSP-1000 | | TSP-10000 | |
|---|---|---|---|---|---|---|---|
| | | GAP↓ | DEGRADATION↓ | GAP↓ | DEGRADATION↓ | GAP↓ | DEGRADATION↓ |
| DIMES | ORI. | 0.69% | 0.06% | 1.11% | 0.07% | 3.05% | 0.00% |
| | GEN. | 0.75% | | 1.18% | | 3.05% | |
| DIFUSCO | ORI. | 0.33% | 0.63% | 0.53% | 2.81% | 2.36% | 0.00% |
| | GEN. | 0.95% | | 3.34% | | 2.36% | |
| SOFTDIST | ORI. | 0.43% | 0.22% | 0.80% | 73.44% | 2.94% | 0.00% |
| | GEN. | 0.65% | | 74.24% | | 2.94% | |
| FAST-T2T | ORI. | 0.12% | 0.23% | 0.65% | 0.06% | 4.22% | 0.00% |
| | GEN. | 0.35% | | 0.71% | | 4.22% | |
| GT-PRIOR | ORI. | 0.50% | 0.01% | 0.85% | 0.04% | 2.13% | 0.00% |
| | GEN. | 0.51% | | 0.89% | | 2.13% | |

## G   ADDITIONAL RESULTS ON TSPLIB

We categorize all Euclidean 2D TSP instances into three groups based on the number of nodes: Small (0-500 nodes), Medium (500-2000 nodes), and Large (more than 2000 nodes). For each category, we evaluate all baseline methods alongside our proposed GT-Prior.

We conducted MCTS evaluations under three distinct parameter settings: (1) Tuned Settings, optimized using uniform TSP instances as listed in Table 14, whose results are shown in Table 9, (2) the Default Settings, as originally employed by Fu et al. (2021), whose results are shown in Table 10, and (3) the Grid Search setting where the MCTS hyperparameters are obtained by instance-level grid search, whose results are shown in Table 11. The results in these tables showcase the performance of the methods in terms of solution length and optimality gap, highlighting the effectiveness of the proposed GT-Prior approach.

The data in Table 8, particularly under the 'Tuned on TSPLIB' setting, further emphasizes the critical role of hyperparameter selection tailored to the specific data distribution. Here, our proposed

Table 8: Generalization performance testing of different methods on TSPLIB instances. The best results in the row are shown in bold and the second-best underlined.

| Size | MCTS Setting | Zero | Att-GCN | DIMES | UTSP | SoftDist | DIFUSCO | Fast-T2T | GT-Prior |
|---|---|---|---|---|---|---|---|---|---|
| Small | Tuned on TSPLIB | 0.06% | **0.05%** | 0.08% | 0.10% | 0.06% | 0.07% | 0.05% | 0.05% |
| | Tuned on Uniform | 0.79% | 0.67% | 0.48% | 0.45% | 1.23% | 0.58% | **0.10%** | 0.77% |
| | Default | 0.87% | 0.67% | 16.01% | 7.16% | 1.80% | 1.06% | 0.96% | **0.20%** |
| Medium | Tuned on TSPLIB | 1.18% | 0.76% | 0.97% | 1.54% | 0.74% | **0.35%** | 0.46% | 0.55% |
| | Tuned on Uniform | 15.24% | 11.47% | 10.64% | 12.03% | 6.79% | **2.38%** | 4.18% | 10.08% |
| | Default | 4.89% | 3.73% | 4.06% | 13.58% | 7.20% | **2.87%** | 5.82% | 3.63% |
| Large | Tuned on TSPLIB | 5.39% | 3.58% | 4.55% | 5.75% | 3.03% | 3.47% | 3.23% | **2.42%** |
| | Tuned on Uniform | 5.54% | 3.92% | 5.48% | 26.51% | 4.43% | 5.68% | 6.81% | **3.53%** |
| | Default | 6.51% | **4.84%** | 391.89% | 1481.66% | 11.36% | 12.62% | 25.77% | 5.51% |

GT-Prior method consistently demonstrates strong performance, achieving the leading optimality gap for Large instances (2.42%) and tying for the best on Small instances (0.05%). Similarly, other approaches like Att-GCN (0.05% on Small) and DIFUSCO (0.35% on Medium) also exhibit their most competitive results when tuned directly on TSPLIB. This underscores that substantial performance gains are unlocked when hyperparameters align with the problem's characteristics. While more granular instance-level hyperparameter optimization, such as the aforementioned grid search (detailed in Table 11), can yield further benefits, its current computational demands are considerable. Consequently, the efficacy of targeted tuning observed in Table 8 strongly motivates future research into efficient hyperparameter optimization, including the development of recommendation systems that could predict near-optimal settings from instance features, thereby achieving robust performance without exhaustive search.

Several key insights emerge from detailed experimental results. First, we observe a strong interaction between instance distribution and parameter tuning effectiveness. While methods like UTSP and DIMES excel on small uniform instances, their performance exhibits high sensitivity to parameter settings when faced with real-world TSPLIB instances, particularly at larger scales (e.g., UTSP degrading from 26.51% to 1481.66% on large instances). This finding reveals a fundamental generalization challenge shared by most learning-based methods - the optimal parameters learned from one distribution may not transfer effectively to another, highlighting the critical importance of robust parameter tuning strategies. To illustrate this distribution sensitivity, we visualize representative hard and easy instances from each group in Figure 6, demonstrating that hard instances deviate significantly from uniform distribution while easy instances closely resemble it.

This generalization issue is particularly noteworthy as it affects all methods except the Zero heatmap, which maintains relatively stable performance across different instance sizes and parameter settings. The Zero heatmap's consistency (varying only from 5.54% to 6.51% on large instances) provides compelling evidence for our thesis that the MCTS component's contribution to solution quality has been historically undervalued in the framework. Furthermore, this stability suggests that proper MCTS parameter tuning might be more crucial for achieving robust performance than developing increasingly sophisticated heatmap generation methods.

From a practical perspective, our analysis also reveals an important computational consideration. The learning-based baselines necessitate GPU resources for both training and inference stages, potentially creating a bottleneck when dealing with real-world data. In contrast, methods that reduce reliance on complex learned components might offer more practical utility in resource-constrained settings while maintaining competitive performance through careful parameter optimization.

These findings collectively suggest that future research in this domain might benefit from a more balanced focus between heatmap sophistication and MCTS optimization, particularly when considering real-world applications where robustness and computational efficiency are paramount.

## H    TUNED HYPERPARAMETER SETTINGS

In this section, we present the search space in Table 12 and results of hyperparameter tuning, summarized in the following Table 14. The table includes the various hyperparameter combinations explored during the tuning process and their corresponding heatmap generation methods.

Table 9: Performance of different methods on TSPLIB instances of varying sizes. All hyperparameter settings are tuned on uniform TSP instances as listed in Table 14.

(a) Small instances (0–500 nodes)

| Instance | Optimal | Zero Length ↓ | Zero Gap ↓ | Att-GCN Length ↓ | Att-GCN Gap ↓ | DIMES Length ↓ | DIMES Gap ↓ | UTSP Length ↓ | UTSP Gap ↓ | SoftDist Length ↓ | SoftDist Gap ↓ | DIFUSCO Length ↓ | DIFUSCO Gap ↓ | Fast-T2T Length ↓ | Fast-T2T Gap ↓ | GT-Prior Length ↓ | GT-Prior Gap ↓ |
|---|---|---|---|---|---|---|---|---|---|---|---|---|---|---|---|---|---|
| st70 | 675 | **676** | 0.15% | **676** | 0.15% | **676** | 0.15% | **676** | 0.15% | 724 | 7.26% | **676** | 0.15% | 676 | 0.15% | 676 | 0.15% |
| eil76 | 538 | **538** | 0.00% | 538 | 0.00% | 538 | 0.00% | 538 | 0.00% | 538 | 0.00% | 538 | 0.00% | 538 | 0.00% | 538 | 0.00% |
| kroA200 | 29368 | **29368** | 0.00% | 29383 | 0.05% | **29368** | 0.00% | 29382 | 0.05% | 29383 | 0.05% | 29380 | 0.04% | 29383 | 0.05% | **29368** | 0.00% |
| eil51 | 426 | **427** | 0.23% | 427 | 0.23% | 427 | 0.23% | 427 | 0.23% | 427 | 0.23% | 427 | 0.23% | 427 | 0.23% | 427 | 0.23% |
| rat195 | 2323 | 2328 | 0.22% | 2328 | 0.22% | **2323** | 0.00% | 2328 | 0.22% | 2328 | 0.22% | 2328 | 0.22% | 2325 | 0.09% | 2328 | 0.22% |
| pr144 | 58537 | 59932 | 2.38% | 63736 | 8.88% | 59553 | 1.74% | 59211 | 1.15% | 66950 | 14.37% | 63389 | 8.29% | **58537** | 0.00% | 65486 | 11.87% |
| bier127 | 118282 | **118282** | 0.00% | **118282** | 0.00% | **118282** | 0.00% | **118282** | 0.00% | **118282** | 0.00% | **118282** | 0.00% | **118282** | 0.00% | **118282** | 0.00% |
| lin105 | 14379 | **14379** | 0.00% | **14379** | 0.00% | **14379** | 0.00% | **14379** | 0.00% | 15081 | 4.88% | **14379** | 0.00% | **14379** | 0.00% | **14379** | 0.00% |
| kroD100 | 21294 | **21294** | 0.00% | **21294** | 0.00% | **21294** | 0.00% | **21294** | 0.00% | **21294** | 0.00% | **21294** | 0.00% | **21294** | 0.00% | **21294** | 0.00% |
| kroA100 | 21282 | **21282** | 0.00% | **21282** | 0.00% | **21282** | 0.00% | **21282** | 0.00% | **21282** | 0.00% | **21282** | 0.00% | **21282** | 0.00% | **21282** | 0.00% |
| pr152 | 73682 | 73682 | 0.55% | **73682** | 0.00% | **73682** | 0.00% | 73818 | 0.18% | 74443 | 1.03% | 74609 | 1.26% | 73900 | 0.30% | 74274 | 0.80% |
| ts225 | 126643 | **126643** | 0.00% | **126643** | 0.00% | **126643** | 0.00% | **126643** | 0.00% | **126643** | 0.00% | **126643** | 0.00% | **126643** | 0.00% | **126643** | 0.00% |
| rd400 | 15281 | 15314 | 0.22% | 15333 | 0.34% | 15323 | 0.27% | 15408 | 0.83% | 15352 | 0.46% | 15320 | 0.26% | **15281** | 0.00% | 15303 | 0.14% |
| kroB100 | 22141 | **22141** | 0.00% | **22141** | 0.00% | **22141** | 0.00% | **22141** | 0.00% | **22141** | 0.00% | **22141** | 0.00% | **22141** | 0.00% | **22141** | 0.00% |
| d198 | 15780 | 15817 | 0.23% | 16344 | 3.57% | 15844 | 0.41% | 15804 | 0.15% | 15816 | 0.23% | 16237 | 2.90% | **15782** | 0.01% | 15817 | 0.23% |
| eil101 | 629 | **629** | 0.00% | **629** | 0.00% | **629** | 0.00% | **629** | 0.00% | **629** | 0.00% | **629** | 0.00% | **629** | 0.00% | **629** | 0.00% |
| linhp318 | 41345 | 42558 | 2.93% | 42523 | 2.85% | 42763 | 3.43% | 42420 | 2.60% | 42283 | 2.27% | 42223 | 2.12% | **42083** | 1.78% | 42387 | 2.52% |
| gil262 | 2378 | 2380 | 0.08% | 2382 | 0.17% | 2380 | 0.08% | 2380 | 0.08% | **2379** | 0.04% | 2380 | 0.08% | 2380 | 0.08% | 2380 | 0.08% |
| rat99 | 1211 | **1211** | 0.00% | **1211** | 0.00% | **1211** | 0.00% | **1211** | 0.00% | **1211** | 0.00% | **1211** | 0.00% | **1211** | 0.00% | **1211** | 0.00% |
| berlin52 | 7542 | **7542** | 0.00% | **7542** | 0.00% | **7542** | 0.00% | **7542** | 0.00% | **7542** | 0.00% | **7542** | 0.00% | **7542** | 0.00% | **7542** | 0.00% |
| kroC100 | 20749 | **20749** | 0.00% | **20749** | 0.00% | **20749** | 0.00% | **20749** | 0.00% | **20749** | 0.00% | **20749** | 0.00% | **20749** | 0.00% | **20749** | 0.00% |
| pr226 | 80369 | 87311 | 8.64% | 83828 | 4.30% | 83828 | 4.30% | 81058 | 0.86% | 80850 | 0.60% | 80463 | 0.12% | **80369** | 0.00% | 85793 | 6.75% |
| fl417 | 11861 | 12852 | 8.36% | 11945 | 0.71% | 12169 | 2.60% | 12800 | 7.92% | 13198 | 11.27% | 12158 | 2.50% | 11932 | 0.60% | 12437 | 4.86% |
| kroE100 | 22068 | **22068** | 0.00% | **22068** | 0.00% | **22068** | 0.00% | **22068** | 0.00% | **22068** | 0.00% | **22068** | 0.00% | **22068** | 0.00% | **22068** | 0.00% |
| pr76 | 108159 | **108159** | 0.00% | **108159** | 0.00% | **108159** | 0.00% | **108159** | 0.00% | 109325 | 1.08% | **108159** | 0.00% | **108159** | 0.00% | **108159** | 0.00% |
| ch130 | 6110 | 6111 | 0.02% | 6111 | 0.02% | 6111 | 0.02% | 6111 | 0.02% | 6242 | 2.16% | 6111 | 0.02% | 6111 | 0.02% | 6111 | 0.02% |
| tsp225 | 3916 | 3932 | 0.41% | **3916** | 0.00% | 3919 | 0.08% | 3923 | 0.18% | **3916** | 0.00% | **3916** | 0.00% | **3916** | 0.00% | 3923 | 0.18% |
| rd100 | 7910 | **7910** | 0.00% | **7910** | 0.00% | **7910** | 0.00% | **7910** | 0.00% | 7938 | 0.35% | **7910** | 0.00% | **7910** | 0.00% | **7910** | 0.00% |
| pr264 | 49135 | 51267 | 4.34% | 50451 | 2.68% | 49949 | 1.66% | 49635 | 1.02% | 49374 | 0.49% | 50389 | 2.55% | **49135** | 0.00% | 49508 | 0.76% |
| pr124 | 59030 | 59168 | 0.23% | 59210 | 0.30% | 59551 | 0.88% | 59210 | 0.30% | 59257 | 0.38% | 59688 | 1.11% | 59076 | 0.08% | **59030** | 0.00% |
| kroA150 | 26524 | **26525** | 0.00% | **26525** | 0.00% | **26525** | 0.00% | **26525** | 0.00% | **26525** | 0.00% | **26525** | 0.00% | **26525** | 0.00% | **26525** | 0.00% |
| kroB200 | 29437 | **29437** | 0.00% | 29438 | 0.00% | **29437** | 0.00% | 29446 | 0.03% | **29437** | 0.00% | **29437** | 0.00% | 29452 | 0.05% | **29437** | 0.00% |
| kroB150 | 26130 | 26178 | 0.18% | 26141 | 0.04% | 26176 | 0.18% | 26136 | 0.02% | **26130** | 0.00% | 26143 | 0.05% | **26130** | 0.00% | **26130** | 0.00% |
| pr107 | 44303 | **44303** | 0.00% | 44387 | 0.19% | **44303** | 0.00% | **44303** | 0.00% | **44303** | 0.00% | **44303** | 0.00% | **44303** | 0.00% | **44303** | 0.00% |
| lin318 | 42029 | 42558 | 1.26% | 42561 | 1.27% | 42609 | 1.38% | 42420 | 0.93% | 42283 | 0.60% | 42254 | 0.54% | **42107** | 0.19% | 42387 | 0.85% |
| pr136 | 96772 | **96772** | 0.00% | **96772** | 0.00% | **96772** | 0.00% | **96772** | 0.00% | **96772** | 0.00% | **96772** | 0.00% | **96772** | 0.00% | **96772** | 0.00% |
| pr299 | 48191 | 48279 | 0.18% | 48223 | 0.07% | 48230 | 0.08% | **48191** | 0.00% | 48197 | 0.01% | 48229 | 0.16% | 48223 | 0.07% | 48197 | 0.01% |
| u159 | 42080 | **42080** | 0.00% | **42080** | 0.00% | **42080** | 0.00% | **42080** | 0.00% | 42396 | 0.75% | **42080** | 0.00% | **42080** | 0.00% | **42080** | 0.00% |
| a280 | 2579 | **2579** | 0.00% | **2579** | 0.00% | **2579** | 0.00% | **2579** | 0.00% | **2579** | 0.00% | **2579** | 0.00% | **2579** | 0.00% | **2579** | 0.00% |
| pr439 | 107217 | 109241 | 1.89% | 108944 | 1.61% | 109594 | 2.22% | 108476 | 1.17% | 110701 | 3.25% | 108485 | 1.18% | **107228** | 0.01% | 109624 | 2.24% |
| ch150 | 6528 | **6528** | 0.00% | **6528** | 0.00% | **6528** | 0.00% | 6533 | 0.08% | **6528** | 0.00% | **6528** | 0.00% | **6528** | 0.00% | **6528** | 0.00% |
| d493 | 35002 | 35347 | 0.99% | 35331 | 0.94% | 35318 | 0.90% | 35235 | 0.67% | 35297 | 0.84% | 35292 | 0.83% | **35045** | 0.12% | 35244 | 0.69% |
| pcb442 | 50778 | 50935 | 0.31% | 50902 | 0.24% | 50856 | 0.15% | 51060 | 0.56% | **50847** | 0.14% | 50908 | 0.26% | 51020 | 0.48% | 50927 | 0.29% |
| Average | - | 35281 | 0.79% | 35244 | 0.67% | 35155 | 0.48% | 35050 | 0.45% | 35340 | 1.23% | 35165 | 0.58% | **34933** | 0.10% | 35321 | 0.77% |

(b) Medium instances (500–2000 nodes)

| Instance | Optimal | Zero Length ↓ | Zero Gap ↓ | Att-GCN Length ↓ | Att-GCN Gap ↓ | DIMES Length ↓ | DIMES Gap ↓ | UTSP Length ↓ | UTSP Gap ↓ | SoftDist Length ↓ | SoftDist Gap ↓ | DIFUSCO Length ↓ | DIFUSCO Gap ↓ | Fast-T2T Length ↓ | Fast-T2T Gap ↓ | GT-Prior Length ↓ | GT-Prior Gap ↓ |
|---|---|---|---|---|---|---|---|---|---|---|---|---|---|---|---|---|---|
| u574 | 36905 | 37211 | 0.83% | 37226 | 0.87% | 37399 | 1.34% | 37211 | 0.83% | 37142 | 0.64% | **36989** | 0.23% | 37610 | 1.91% | 37146 | 0.65% |
| pcb1173 | 56892 | 57837 | 1.66% | 57715 | 1.45% | 57618 | 1.28% | 57770 | 1.54% | 57633 | 1.30% | 57304 | 0.72% | **57065** | 0.30% | 57248 | 0.63% |
| rat783 | 8806 | 8903 | 1.10% | 8887 | 0.92% | 8892 | 0.98% | 8919 | 1.28% | 8884 | 0.89% | **8843** | 0.41% | 8875 | 0.78% | 8851 | 0.51% |
| u1432 | 152970 | 156669 | 2.42% | 154684 | 1.12% | 154889 | 1.25% | 154770 | 1.13% | 154338 | 0.89% | **154046** | 0.70% | 155344 | 1.55% | 154285 | 0.86% |
| fl1400 | 20127 | 27446 | 36.36% | 26280 | 30.57% | 23066 | 14.60% | 23467 | 16.59% | 29343 | 45.79% | **21519** | 6.92% | 29367 | 45.91% | 22924 | 13.90% |
| vm1084 | 239297 | 255009 | 6.57% | 257899 | 7.77% | 254512 | 6.36% | 246531 | 3.02% | **240016** | 0.30% | 240265 | 0.40% | 240087 | 0.33% | 244968 | 2.37% |
| rat575 | 6773 | 6844 | 1.05% | 6826 | 0.78% | 6845 | 1.06% | 6829 | 0.83% | 6814 | 0.61% | **6800** | 0.40% | 6854 | 1.20% | 6807 | 0.50% |
| vm1748 | 336556 | 377814 | 12.26% | 385587 | 14.57% | 378032 | 12.32% | 376601 | 11.90% | 341506 | 1.47% | 341443 | 1.45% | **338879** | 0.69% | 343834 | 2.16% |
| rl1889 | 316536 | 479282 | 51.41% | 444184 | 40.33% | 397609 | 25.61% | 441143 | 39.37% | 327774 | 3.55% | **324242** | 2.43% | 330678 | 4.47% | 451948 | 42.78% |
| u724 | 41910 | 42288 | 0.90% | 42105 | 0.47% | 42330 | 1.00% | 42317 | 0.97% | 42161 | 0.60% | **42003** | 0.22% | 42399 | 1.17% | 42045 | 0.42% |
| d1291 | 50801 | 72786 | 43.28% | 70051 | 37.89% | 71972 | 41.67% | 72779 | 43.26% | 52023 | 2.41% | **51342** | 1.06% | 51897 | 2.16% | 74911 | 47.46% |
| pr1002 | 259045 | 265784 | 2.60% | 265316 | 2.43% | 263164 | 1.59% | 264061 | 1.94% | 262591 | 1.37% | 262472 | 1.32% | **261306** | 0.87% | 262929 | 1.50% |
| fl1577 | 22249 | 29723 | 33.59% | 27605 | 24.07% | 30050 | 35.06% | 29581 | 32.95% | 29102 | 30.80% | 25960 | 16.68% | **25099** | 12.81% | 29222 | 31.34% |
| nrw1379 | 56638 | 57171 | 0.94% | 57070 | 0.76% | 57326 | 1.21% | 57172 | 0.94% | 58266 | 2.87% | **56961** | 0.57% | 57328 | 1.22% | 56974 | 0.59% |
| rl1304 | 252948 | 332691 | 31.53% | 316879 | 25.27% | 316925 | 25.29% | 316283 | 25.04% | 262598 | 3.82% | 257797 | 1.92% | **257053** | 1.62% | 297448 | 17.59% |
| d657 | 48912 | 49228 | 0.65% | 49303 | 0.80% | 49350 | 0.90% | 49350 | 0.90% | 49094 | 0.37% | **49008** | 0.38% | 49138 | 0.46% | 49118 | 0.42% |
| p654 | 34643 | 38112 | 10.01% | 38864 | 12.18% | 35210 | 1.64% | 35884 | 3.58% | 47033 | 35.76% | 36765 | 6.13% | **34810** | 0.48% | 35569 | 2.67% |
| d1655 | 62128 | 66466 | 6.98% | 65547 | 5.50% | 64743 | 4.21% | 65977 | 6.20% | 63986 | 2.99% | 64358 | 3.59% | 64434 | 3.71% | **63951** | 2.93% |
| u1817 | 57201 | 90599 | 58.39% | 68245 | 19.31% | 71276 | 24.61% | 80609 | 40.92% | 58838 | 2.86% | **58587** | 2.42% | 59645 | 4.27% | 75131 | 31.35% |
| u1060 | 224094 | 233417 | 4.16% | 232573 | 3.78% | 242781 | 8.34% | 236866 | 5.70% | 227830 | 1.67% | **225164** | 0.48% | 225676 | 0.71% | 229725 | 2.51% |
| rl1323 | 270199 | 306164 | 13.31% | 297453 | 10.09% | 305970 | 13.24% | 307474 | 13.80% | 274440 | 1.57% | 274104 | 1.45% | **273587** | 1.25% | 293294 | 8.55% |
| Average | - | 142449 | 15.24% | 138583 | 11.47% | 136662 | 10.64% | 138644 | 12.03% | 125305 | 6.79% | **123621** | 2.38% | 124149 | 4.18% | 135160 | 10.08% |

(c) Large instances (2000+ nodes)

| Instance | Optimal | Zero Length ↓ | Zero Gap ↓ | Att-GCN Length ↓ | Att-GCN Gap ↓ | DIMES Length ↓ | DIMES Gap ↓ | UTSP Length ↓ | UTSP Gap ↓ | SoftDist Length ↓ | SoftDist Gap ↓ | DIFUSCO Length ↓ | DIFUSCO Gap ↓ | Fast-T2T Length ↓ | Fast-T2T Gap ↓ | GT-Prior Length ↓ | GT-Prior Gap ↓ |
|---|---|---|---|---|---|---|---|---|---|---|---|---|---|---|---|---|---|
| u2152 | 64253 | 66719 | 3.84% | 66301 | 3.19% | 67244 | 4.66% | 79556 | 23.82% | 66354 | 3.27% | 66111 | 2.89% | 67539 | 5.11% | **65467** | 1.89% |
| u2319 | 234256 | 240657 | 2.73% | 236054 | 0.77% | 237061 | 1.20% | 235667 | 0.60% | **234765** | 0.22% | 236201 | 0.83% | 236769 | 1.07% | 235093 | 0.36% |
| pcb3038 | 137694 | 142320 | 3.36% | 141418 | 2.70% | 142646 | 3.60% | 140351 | 1.93% | 139547 | 1.35% | 141446 | 2.72% | 143730 | 4.38% | **139325** | 1.18% |
| fl3795 | 28772 | 35138 | 22.13% | **33971** | 18.07% | 36294 | 26.14% | 43940 | 52.72% | 36803 | 27.91% | 40183 | 39.66% | 38421 | 33.54% | 35715 | 24.13% |
| pr2392 | 378032 | 384727 | 1.77% | 388518 | 2.77% | 386985 | 2.37% | 385057 | 1.86% | 385073 | 1.86% | 387623 | 2.54% | 386244 | 2.17% | **380722** | 0.71% |
| fnl4461 | 182566 | 187380 | 2.64% | 186985 | 2.42% | 187913 | 2.93% | 185869 | 1.81% | **184057** | 0.82% | 186521 | 2.17% | 189064 | 3.56% | 184776 | 1.21% |
| d2103 | 80450 | 83622 | 3.94% | 82614 | 2.69% | 83690 | 4.03% | 86119 | 7.05% | 83644 | 3.97% | 83360 | 3.62% | 84177 | 4.63% | **81813** | 1.69% |
| rl5934 | 556045 | 588550 | 5.85% | 579206 | 4.17% | 589806 | 6.07% | 843158 | 51.63% | **570853** | 2.66% | 594357 | 6.89% | 606923 | 9.15% | 574556 | 3.33% |
| rl5915 | 565530 | 589372 | 4.22% | 588542 | 4.07% | 585404 | 3.51% | 809375 | 43.12% | **578232** | 2.25% | 584327 | 3.32% | 602911 | 6.61% | 583477 | 3.17% |
| usa13509 | 19982859 | 20947758 | 4.83% | 20613997 | 3.16% | 21033416 | 5.26% | 28386893 | 42.06% | 21193246 | 6.06% | 20723480 | 3.71% | 21075706 | 5.47% | **20396752** | 2.07% |
| brd14051 | 469385 | 492159 | 4.85% | 480186 | 2.30% | 489324 | 4.25% | 506961 | 8.01% | 485812 | 3.50% | 482790 | 2.86% | 493303 | 5.10% | **479123** | 2.07% |
| d18512 | 645238 | 672990 | 4.30% | 662312 | 2.65% | 667466 | 3.44% | 701169 | 8.67% | 663460 | 2.82% | 662022 | 2.60% | 672430 | 4.21% | **656164** | 1.69% |
| rl11849 | 923288 | 994084 | 7.67% | 955040 | 3.44% | 973842 | 5.48% | 1866653 | 102.17% | **948548** | 2.74% | 953754 | 3.30% | 980279 | 6.17% | 962460 | 4.24% |
| d15112 | 1573084 | 1659366 | 5.48% | 1613134 | 2.55% | 1631994 | 3.74% | 1978136 | 25.75% | 1614098 | 2.61% | 1612163 | 2.48% | 1638424 | 4.15% | **1598467** | 1.61% |
| Average | - | 1934631 | 5.54% | 1902019 | 3.92% | 1936648 | 5.48% | 2589207 | 26.51% | 1941749 | 4.43% | 1911024 | 5.68% | 1943994 | 6.81% | **1883850** | 3.53% |

Table 10: Performance of different methods on TSPLIB instances of varying sizes. The hyperparameter settings are the default settings as used by Fu et al. (2021).

(a) Small TSPLIB instances (0–500 nodes)

| Instance | Optimal | Zero Length↓ | Zero Gap↓ | Att-GCN Length↓ | Att-GCN Gap↓ | DIMES Length↓ | DIMES Gap↓ | UTSP Length↓ | UTSP Gap↓ | SoftDist Length↓ | SoftDist Gap↓ | DIFUSCO Length↓ | DIFUSCO Gap↓ | Fast-T2T Length↓ | Fast-T2T Gap↓ | GT-Prior Length↓ | GT-Prior Gap↓ |
|---|---|---|---|---|---|---|---|---|---|---|---|---|---|---|---|---|---|
| st70 | 675 | 676 | 0.15% | 676 | 0.15% | 1056 | 56.44% | 676 | 0.15% | 676 | 0.15% | 694 | 2.81% | 679 | 0.59% | 676 | 0.15% |
| kroA200 | 29368 | 29635 | 0.91% | 29368 | 0.00% | 29464 | 0.33% | 29529 | 0.55% | 29383 | 0.05% | 29831 | 1.58% | 29427 | 0.20% | 29397 | 0.10% |
| eil76 | 538 | 538 | 0.00% | 538 | 0.00% | 803 | 49.26% | 538 | 0.00% | 538 | 0.00% | 538 | 1.15% | 538 | 0.00% | 538 | 0.00% |
| pr144 | 58537 | 58554 | 0.03% | 67632 | 15.54% | 72458 | 23.78% | 58537 | 0.00% | 66184 | 13.06% | 58901 | 0.62% | 58640 | 0.18% | 58537 | 0.00% |
| rat195 | 2323 | 2365 | 1.81% | 2323 | 0.00% | 2331 | 0.34% | 2352 | 1.25% | 2323 | 0.00% | 2337 | 0.60% | 2334 | 0.47% | 2328 | 0.22% |
| eil51 | 426 | 427 | 0.23% | 427 | 0.23% | 653 | 53.29% | 427 | 0.23% | 427 | 0.23% | 427 | 0.23% | 427 | 0.23% | 427 | 0.23% |
| bier127 | 118282 | 118580 | 0.25% | 118282 | 0.00% | 118715 | 0.37% | 118282 | 0.00% | 118423 | 0.12% | 118657 | 0.32% | 118385 | 0.09% | 118282 | 0.00% |
| lin105 | 14379 | 14379 | 0.00% | 14379 | 0.00% | 16437 | 14.31% | 14379 | 0.00% | 15073 | 4.83% | 14401 | 0.15% | 14401 | 0.15% | 14379 | 0.00% |
| kroD100 | 21294 | 21294 | 0.00% | 21294 | 0.00% | 28391 | 33.33% | 21309 | 0.07% | 21294 | 0.00% | 21374 | 0.38% | 21294 | 0.00% | 21294 | 0.00% |
| pr152 | 73682 | 73880 | 0.27% | 73682 | 0.00% | 86257 | 17.07% | 73682 | 0.00% | 73682 | 0.00% | 74029 | 0.47% | 74121 | 0.60% | 73682 | 0.00% |
| kroA100 | 21282 | 21282 | 0.00% | 21282 | 0.00% | 25168 | 18.26% | 21282 | 0.00% | 21282 | 0.00% | 21396 | 0.54% | 21768 | 2.28% | 21282 | 0.00% |
| ts225 | 126643 | 127147 | 0.40% | 126713 | 0.06% | 143360 | 13.20% | 126726 | 0.07% | 126962 | 0.25% | 126643 | 0.00% | 127104 | 0.36% | 126643 | 0.00% |
| rd400 | 15281 | 15819 | 3.52% | 15413 | 0.86% | 15829 | 3.59% | 15580 | 1.96% | 15418 | 0.90% | 15350 | 0.45% | 15313 | 0.21% | 15454 | 1.13% |
| kroB100 | 22141 | 22193 | 0.23% | 22141 | 0.00% | 26014 | 17.49% | 22141 | 0.00% | 22141 | 0.00% | 22601 | 2.08% | 22858 | 3.24% | 22141 | 0.00% |
| d198 | 15780 | 15883 | 0.65% | 15784 | 0.03% | 16016 | 1.50% | 15874 | 0.60% | 15806 | 0.16% | 15859 | 0.50% | 16226 | 2.83% | 15789 | 0.06% |
| eil101 | 629 | 630 | 0.16% | 629 | 0.00% | 914 | 45.31% | 629 | 0.00% | 629 | 0.00% | 629 | 0.00% | 629 | 0.00% | 629 | 0.00% |
| linhp318 | 41345 | 43250 | 4.61% | 42359 | 2.45% | 43263 | 4.64% | 42453 | 2.68% | 43111 | 4.27% | 42336 | 2.40% | 42734 | 3.36% | 42212 | 2.10% |
| gil262 | 2378 | 2433 | 2.31% | 2383 | 0.21% | 2482 | 4.37% | 2394 | 0.67% | 2392 | 0.59% | 2380 | 0.08% | 2380 | 0.08% | 2389 | 0.46% |
| rat99 | 1211 | 1211 | 0.00% | 1211 | 0.00% | 1218 | 0.58% | 1211 | 0.00% | 1211 | 0.00% | 1214 | 0.25% | 1214 | 0.25% | 1211 | 0.00% |
| berlin52 | 7542 | 7542 | 0.00% | 7542 | 0.00% | 10569 | 40.14% | 7542 | 0.00% | 7542 | 0.00% | 7542 | 0.00% | 7542 | 0.00% | 7542 | 0.00% |
| kroC100 | 20749 | 20749 | 0.00% | 20749 | 0.00% | 24666 | 18.88% | 20749 | 0.00% | 20749 | 0.00% | 20901 | 0.73% | 21177 | 2.06% | 20749 | 0.00% |
| pr226 | 80369 | 80822 | 0.56% | 83203 | 3.53% | 84543 | 5.19% | 81060 | 0.86% | 85411 | 6.27% | 83028 | 3.31% | 80518 | 0.19% | 80369 | 0.00% |
| fl417 | 11861 | 11932 | 0.60% | 12014 | 1.29% | 14036 | 18.34% | 45810 | 286.22% | 14897 | 25.60% | 13977 | 17.84% | 12791 | 7.84% | 11907 | 0.39% |
| kroE100 | 22068 | 22068 | 0.00% | 22068 | 0.00% | 26062 | 18.10% | 22068 | 0.00% | 22068 | 0.00% | 22135 | 0.30% | 22121 | 0.24% | 22068 | 0.00% |
| pr76 | 108159 | 108159 | 0.00% | 108159 | 0.00% | 130741 | 20.88% | 108159 | 0.00% | 109325 | 1.08% | 111683 | 3.26% | 109838 | 1.55% | 108159 | 0.00% |
| ch130 | 6110 | 6149 | 0.64% | 6111 | 0.02% | 7706 | 26.12% | 6120 | 0.16% | 6248 | 2.26% | 6157 | 0.77% | 6129 | 0.31% | 6111 | 0.02% |
| rd100 | 7910 | 7910 | 0.00% | 7910 | 0.00% | 14528 | 83.67% | 7910 | 0.00% | 7932 | 0.28% | 7910 | 0.00% | 7910 | 0.00% | 7910 | 0.00% |
| tsp225 | 3916 | 3982 | 1.69% | 3923 | 0.18% | 3945 | 0.74% | 3966 | 1.28% | 3919 | 0.08% | 3920 | 0.10% | 4026 | 2.81% | 3923 | 0.18% |
| pr264 | 49135 | 49552 | 0.85% | 49135 | 0.00% | 49248 | 0.23% | 49844 | 1.44% | 49309 | 0.35% | 49180 | 0.09% | 49135 | 0.00% | 49135 | 0.00% |
| pr124 | 59030 | 59030 | 0.00% | 59030 | 0.00% | 76615 | 29.79% | 59030 | 0.00% | 59524 | 0.84% | 59385 | 0.60% | 59159 | 0.22% | 59030 | 0.00% |
| kroA150 | 26524 | 26726 | 0.76% | 26525 | 0.00% | 26719 | 0.74% | 26528 | 0.02% | 26525 | 0.00% | 26556 | 0.12% | 26829 | 1.15% | 26525 | 0.00% |
| kroB200 | 29437 | 29619 | 0.62% | 29455 | 0.06% | 29511 | 0.25% | 29552 | 0.39% | 29438 | 0.00% | 29659 | 0.75% | 29726 | 0.98% | 29475 | 0.13% |
| kroB150 | 26130 | 26143 | 0.05% | 26132 | 0.01% | 26335 | 0.78% | 26176 | 0.18% | 26130 | 0.00% | 26149 | 0.07% | 26330 | 0.77% | 26130 | 0.00% |
| pr107 | 44303 | 44358 | 0.12% | 44387 | 0.19% | 48621 | 9.75% | 44303 | 0.00% | 44303 | 0.00% | 44387 | 0.19% | 44347 | 0.10% | 44303 | 0.00% |
| lin318 | 42029 | 43250 | 2.91% | 42352 | 0.77% | 43116 | 2.59% | 42453 | 1.01% | 43111 | 2.57% | 42646 | 1.47% | 42336 | 0.73% | 42212 | 0.44% |
| pr136 | 96772 | 97515 | 0.77% | 96772 | 0.00% | 119314 | 23.29% | 96785 | 0.01% | 96772 | 0.00% | 96781 | 0.01% | 96994 | 0.23% | 96772 | 0.00% |
| pr299 | 48191 | 48979 | 1.64% | 48300 | 0.18% | 48257 | 0.14% | 48594 | 0.84% | 48241 | 0.10% | 48306 | 0.24% | 49731 | 3.20% | 48303 | 0.23% |
| u159 | 42080 | 42080 | 0.00% | 42080 | 0.00% | 43188 | 2.63% | 42080 | 0.00% | 42396 | 0.75% | 42685 | 1.44% | 42080 | 0.00% | 42080 | 0.00% |
| a280 | 2579 | 2633 | 2.09% | 2579 | 0.00% | 2581 | 0.08% | 2589 | 0.39% | 2581 | 0.08% | 2579 | 0.00% | 2579 | 0.00% | 2585 | 0.23% |
| pr439 | 107217 | 109872 | 2.48% | 108631 | 1.32% | 108602 | 1.29% | 108424 | 1.13% | 115530 | 7.75% | 108855 | 1.53% | 107458 | 0.22% | 107656 | 0.41% |
| ch150 | 6528 | 6562 | 0.52% | 6528 | 0.00% | 8178 | 25.28% | 6528 | 0.00% | 6528 | 0.00% | 6533 | 0.08% | 6562 | 0.52% | 6528 | 0.00% |
| d493 | 35002 | 35874 | 2.49% | 35373 | 1.06% | 35522 | 1.49% | 36384 | 3.95% | 35480 | 1.37% | 35537 | 1.53% | 35665 | 1.89% | 35487 | 1.39% |
| pcb442 | 50778 | 52292 | 2.98% | 51098 | 0.63% | 51147 | 0.73% | 51775 | 1.96% | 51177 | 0.79% | 50976 | 0.39% | 51287 | 1.00% | 51095 | 0.62% |
| Average | - | 35208 | 0.87% | 35268 | 0.67% | 38711 | 16.01% | 35870 | 7.16% | 35630 | 1.80% | 35280 | 1.06% | 35180 | 0.96% | 34961 | 0.20% |

(b) Medium TSPLIB instances (500–2000 nodes)

| Instance | Optimal | Zero Length↓ | Zero Gap↓ | Att-GCN Length↓ | Att-GCN Gap↓ | DIMES Length↓ | DIMES Gap↓ | UTSP Length↓ | UTSP Gap↓ | SoftDist Length↓ | SoftDist Gap↓ | DIFUSCO Length↓ | DIFUSCO Gap↓ | Fast-T2T Length↓ | Fast-T2T Gap↓ | GT-Prior Length↓ | GT-Prior Gap↓ |
|---|---|---|---|---|---|---|---|---|---|---|---|---|---|---|---|---|---|
| u574 | 36905 | 38171 | 3.43% | 37545 | 1.73% | 37803 | 2.43% | 38018 | 3.02% | 37545 | 1.73% | 37026 | 0.33% | 37632 | 1.97% | 37449 | 1.45% |
| pcb1173 | 56892 | 60231 | 5.87% | 58452 | 2.74% | 58664 | 3.11% | 59761 | 5.04% | 58209 | 2.31% | 57717 | 1.45% | 57166 | 0.48% | 58251 | 2.39% |
| u1432 | 152970 | 162741 | 6.39% | 157322 | 2.85% | 157056 | 2.67% | 159654 | 4.37% | 155566 | 1.70% | 154734 | 1.15% | 156271 | 2.16% | 156126 | 2.06% |
| rat783 | 8806 | 9230 | 4.81% | 8995 | 2.15% | 9088 | 3.20% | 9124 | 3.61% | 8936 | 1.48% | 8863 | 0.65% | 8888 | 0.93% | 8986 | 2.04% |
| fl1400 | 20127 | 20917 | 3.93% | 23347 | 16.00% | 20932 | 4.00% | 37919 | 88.40% | 30111 | 49.61% | 22608 | 12.33% | 29913 | 48.62% | 21272 | 5.69% |
| vm1084 | 239297 | 251602 | 5.14% | 242848 | 1.48% | 245994 | 2.80% | 252204 | 5.39% | 243541 | 1.77% | 242375 | 1.29% | 242670 | 1.41% | 244267 | 2.08% |
| rat575 | 6773 | 6982 | 3.09% | 6901 | 1.89% | 7053 | 4.13% | 6959 | 2.75% | 6871 | 1.45% | 6801 | 0.41% | 6860 | 1.28% | 6842 | 1.02% |
| vm1748 | 336556 | 352556 | 4.75% | 344077 | 2.23% | 347356 | 3.21% | 372117 | 10.57% | 344193 | 2.27% | 340888 | 1.29% | 349067 | 3.72% | 343973 | 2.20% |
| rl1889 | 316536 | 335641 | 6.04% | 325270 | 2.76% | 338164 | 6.83% | 358570 | 13.28% | 329839 | 4.20% | 322969 | 2.03% | 329813 | 4.19% | 328399 | 3.75% |
| u724 | 41910 | 43487 | 3.76% | 42525 | 1.47% | 42915 | 2.40% | 43106 | 2.85% | 42508 | 1.43% | 42081 | 0.41% | 42452 | 1.29% | 42420 | 1.22% |
| d1291 | 50801 | 52757 | 3.85% | 52063 | 2.48% | 53833 | 5.97% | 54231 | 6.75% | 52230 | 2.81% | 51937 | 2.24% | 53543 | 5.40% | 52553 | 3.45% |
| pr1002 | 259045 | 273143 | 5.44% | 264647 | 2.16% | 267949 | 3.44% | 268931 | 3.82% | 266468 | 2.87% | 263242 | 1.62% | 270563 | 4.45% | 264704 | 2.18% |
| fl1577 | 22249 | 23351 | 4.95% | 26082 | 17.23% | 23954 | 7.66% | 27592 | 24.01% | 28630 | 28.68% | 25493 | 14.58% | 24560 | 10.39% | 27531 | 23.74% |
| nrw1379 | 56638 | 58991 | 4.15% | 57681 | 1.84% | 57737 | 1.94% | 65399 | 15.47% | 58021 | 2.44% | 57297 | 1.16% | 57460 | 1.45% | 57654 | 1.79% |
| rl1304 | 252948 | 270179 | 6.81% | 259681 | 2.66% | 270057 | 6.76% | 268425 | 6.12% | 264884 | 4.72% | 255970 | 1.19% | 263163 | 4.04% | 263748 | 4.27% |
| d657 | 48912 | 50971 | 4.21% | 49798 | 1.81% | 50577 | 3.40% | 50437 | 3.12% | 49657 | 1.52% | 49153 | 0.49% | 49951 | 2.12% | 49616 | 1.44% |
| p654 | 34643 | 35266 | 1.80% | 36233 | 4.59% | 35873 | 3.55% | 49921 | 44.10% | 44016 | 27.06% | 37936 | 9.51% | 37851 | 9.26% | 35979 | 3.86% |
| dl1655 | 62128 | 66819 | 7.55% | 63970 | 2.96% | 64668 | 4.09% | 75875 | 22.13% | 64467 | 3.76% | 63575 | 2.33% | 65308 | 5.12% | 63610 | 2.39% |
| u1817 | 57201 | 61671 | 7.81% | 59226 | 3.54% | 60219 | 5.28% | 63152 | 10.40% | 59585 | 4.17% | 58780 | 2.76% | 62759 | 9.72% | 59318 | 3.70% |
| u1060 | 224094 | 232616 | 3.80% | 227340 | 1.45% | 232619 | 3.80% | 236167 | 5.39% | 228869 | 2.13% | 227868 | 1.68% | 227580 | 1.56% | 229515 | 2.42% |
| rl1323 | 270199 | 283701 | 5.00% | 276363 | 2.28% | 282500 | 4.55% | 282676 | 4.62% | 278379 | 3.03% | 274038 | 1.42% | 277387 | 2.66% | 278283 | 2.99% |
| Average | - | 128143 | 4.89% | 124779 | 3.73% | 126905 | 4.06% | 132392 | 13.58% | 126310 | 7.20% | 123873 | 2.87% | 126231 | 5.82% | 125261 | 3.63% |

(c) Large TSPLIB instances (>2000 nodes)

| Instance | Optimal | Zero Length↓ | Zero Gap↓ | Att-GCN Length↓ | Att-GCN Gap↓ | DIMES Length↓ | DIMES Gap↓ | UTSP Length↓ | UTSP Gap↓ | SoftDist Length↓ | SoftDist Gap↓ | DIFUSCO Length↓ | DIFUSCO Gap↓ | Fast-T2T Length↓ | Fast-T2T Gap↓ | GT-Prior Length↓ | GT-Prior Gap↓ |
|---|---|---|---|---|---|---|---|---|---|---|---|---|---|---|---|---|---|
| u2152 | 64253 | 68293 | 6.29% | 66717 | 3.83% | 69322 | 7.89% | 71240 | 10.87% | 96834 | 50.71% | 77826 | 21.12% | 75822 | 18.01% | 66600 | 3.65% |
| u2319 | 234256 | 243093 | 3.77% | 237114 | 1.22% | 251125 | 7.20% | 244142 | 4.22% | 235644 | 0.59% | 237035 | 1.19% | 237114 | 1.22% | 236159 | 0.81% |
| pcb3038 | 137694 | 150518 | 9.31% | 142015 | 3.14% | 163500 | 18.74% | 148143 | 7.59% | 141977 | 3.11% | 157341 | 14.27% | 156910 | 13.96% | 141372 | 2.67% |
| fl3795 | 28772 | 30032 | 4.38% | 35694 | 24.06% | 35201 | 22.34% | 50835 | 76.68% | 36579 | 27.13% | 42120 | 46.39% | 46458 | 61.47% | 38852 | 35.03% |
| pr2392 | 378032 | 392998 | 3.96% | 391367 | 3.53% | 426194 | 12.74% | 401216 | 6.13% | 438424 | 15.98% | 430218 | 13.80% | 395299 | 4.57% | 385009 | 1.85% |
| fnl4461 | 182566 | 192471 | 5.43% | 187802 | 2.87% | 235876 | 29.20% | 229934 | 25.95% | 186632 | 2.23% | 192868 | 5.64% | 213327 | 16.85% | 186359 | 2.08% |
| d2103 | 80450 | 88698 | 10.25% | 83881 | 4.26% | 96968 | 20.53% | 88022 | 9.41% | 84662 | 5.24% | 90773 | 12.83% | 87161 | 8.34% | 82723 | 2.83% |
| rl5934 | 556045 | 590393 | 6.18% | 576829 | 3.74% | 703750 | 26.56% | 781490 | 40.54% | 647689 | 16.48% | 645291 | 16.05% | 684076 | 23.03% | 592889 | 6.63% |
| rl5915 | 565530 | 603653 | 6.74% | 587231 | 3.84% | 694199 | 22.75% | 809014 | 43.05% | 644676 | 14.00% | 656872 | 16.15% | 717361 | 26.85% | 591517 | 4.60% |
| usa13509 | 19982859 | 21177174 | 5.98% | 20733868 | 3.76% | 442759283 | 2115.70% | 1115269461 | 5481.13% | 21094456 | 5.56% | 22241850 | 11.30% | 22891847 | 14.56% | 20742301 | 3.80% |
| brd14051 | 469385 | 496359 | 5.75% | 484032 | 3.12% | 3757018 | 700.41% | 13600054 | 2797.42% | 493461 | 5.13% | 489311 | 4.25% | 622791 | 32.68% | 483657 | 3.04% |
| d18512 | 645238 | 685983 | 6.31% | 665993 | 3.22% | 4922388 | 662.88% | 22893796 | 3448.12% | 664334 | 2.96% | 663087 | 2.77% | 839806 | 30.15% | 659537 | 2.22% |
| rl11849 | 923288 | 1014118 | 9.84% | 961746 | 4.17% | 7381138 | 699.44% | 40891587 | 4328.91% | 990268 | 7.25% | 977396 | 5.86% | 1421607 | 53.97% | 970070 | 5.07% |
| d15112 | 1573084 | 1681649 | 6.90% | 1621028 | 3.05% | 19507797 | 1140.10% | 71782581 | 4463.18% | 1615421 | 2.69% | 1653223 | 5.09% | 2440649 | 55.15% | 1618636 | 2.90% |
| Average | - | 1958245 | 6.51% | 1912522 | 4.84% | 34357411 | 391.89% | 90518679 | 1481.66% | 1955075 | 11.36% | 2039657 | 12.62% | 2202159 | 25.77% | 1913977 | 5.51% |

Table 11: Performance of different methods on TSPLIB instances of varying sizes. The hyperparameter settings are obtained by grid search.

### (a) Small instances (0–500 nodes)

| Instance | Optimal | Zero | | Att-GCN | | DIMES | | UTSP | | SoftDist | | DIFUSCO | | Fast-T2T | | GT-Prior | |
|---|---|---|---|---|---|---|---|---|---|---|---|---|---|---|---|---|---|
| | | Length ↓ | Gap ↓ | Length ↓ | Gap ↓ | Length ↓ | Gap ↓ | Length ↓ | Gap ↓ | Length ↓ | Gap ↓ | Length ↓ | Gap ↓ | Length ↓ | Gap ↓ | Length ↓ | Gap ↓ |
| st70 | 675 | **675** | **0.00%** | 676 | 0.15% | **675** | **0.00%** | 676 | 0.15% | 676 | 0.15% | 676 | 0.15% | 676 | 0.15% | 676 | 0.15% |
| kroA200 | 29368 | **29368** | **0.00%** | 29368 | 0.00% | **29368** | **0.00%** | 29368 | 0.00% | 29368 | 0.00% | 29368 | 0.00% | 29368 | 0.00% | 29368 | 0.00% |
| eil76 | 538 | **538** | **0.00%** | 538 | 0.00% | 538 | 0.00% | 538 | 0.00% | 538 | 0.00% | 538 | 0.00% | 538 | 0.00% | 538 | 0.00% |
| pr144 | 58537 | **58537** | **0.00%** | 58537 | 0.00% | 58537 | 0.00% | 58537 | 0.00% | 58537 | 0.00% | 58537 | 0.00% | 58537 | 0.00% | 58537 | 0.00% |
| rat195 | 2323 | **2323** | **0.00%** | 2323 | 0.00% | 2323 | 0.00% | 2323 | 0.00% | 2323 | 0.00% | 2323 | 0.00% | 2323 | 0.00% | 2323 | 0.00% |
| eil51 | 426 | **427** | **0.23%** | 427 | 0.23% | 427 | 0.23% | 427 | 0.23% | 427 | 0.23% | 427 | 0.23% | 427 | 0.23% | 427 | 0.23% |
| bier127 | 118282 | **118282** | **0.00%** | 118282 | 0.00% | 118282 | 0.00% | 118282 | 0.00% | 118282 | 0.00% | 118282 | 0.00% | 118282 | 0.00% | 118282 | 0.00% |
| lin105 | 14379 | **14379** | **0.00%** | 14379 | 0.00% | 14379 | 0.00% | 14379 | 0.00% | 14379 | 0.00% | 14379 | 0.00% | 14379 | 0.00% | 14379 | 0.00% |
| kroD100 | 21294 | **21294** | **0.00%** | 21294 | 0.00% | 21294 | 0.00% | 21294 | 0.00% | 21294 | 0.00% | 21294 | 0.00% | 21294 | 0.00% | 21294 | 0.00% |
| pr152 | 73682 | **73682** | **0.00%** | 73682 | 0.00% | 73682 | 0.00% | 73682 | 0.00% | 73682 | 0.00% | 73682 | 0.00% | 73682 | 0.00% | 73682 | 0.00% |
| kroA100 | 21282 | **21282** | **0.00%** | 21282 | 0.00% | 21282 | 0.00% | 21282 | 0.00% | 21282 | 0.00% | 21282 | 0.00% | 21282 | 0.00% | 21282 | 0.00% |
| ts225 | 126643 | **126643** | **0.00%** | 126643 | 0.00% | 126643 | 0.00% | 126643 | 0.00% | 126643 | 0.00% | 126643 | 0.00% | 126643 | 0.00% | 126643 | 0.00% |
| rd400 | 15281 | 15289 | 0.05% | 15295 | 0.09% | 15300 | 0.12% | 15323 | 0.27% | 15291 | 0.07% | 15288 | 0.05% | **15281** | **0.00%** | 15292 | 0.07% |
| kroB100 | 22141 | **22141** | **0.00%** | 22141 | 0.00% | 22141 | 0.00% | 22141 | 0.00% | 22141 | 0.00% | 22141 | 0.00% | 22141 | 0.00% | 22141 | 0.00% |
| d198 | 15780 | **15780** | **0.00%** | 15780 | 0.00% | 15780 | 0.00% | 15794 | 0.09% | 15780 | 0.00% | 15780 | 0.00% | 15780 | 0.00% | 15780 | 0.00% |
| eil101 | 629 | **629** | **0.00%** | 629 | 0.00% | 629 | 0.00% | 629 | 0.00% | 629 | 0.00% | 629 | 0.00% | 629 | 0.00% | 629 | 0.00% |
| linhp318 | 41345 | 42080 | 1.78% | **42029** | **1.65%** | 42175 | 2.01% | 42194 | 2.05% | 42029 | 1.65% | 42029 | 1.65% | 42029 | 1.65% | 42029 | 1.65% |
| gil262 | 2378 | 2379 | 0.04% | 2379 | 0.04% | 2379 | 0.04% | 2379 | 0.04% | 2378 | 0.00% | 2379 | 0.04% | 2379 | 0.04% | 2379 | 0.04% |
| rat99 | 1211 | **1211** | **0.00%** | 1211 | 0.00% | 1211 | 0.00% | 1211 | 0.00% | 1211 | 0.00% | 1211 | 0.00% | 1211 | 0.00% | 1211 | 0.00% |
| berlin52 | 7542 | **7542** | **0.00%** | 7542 | 0.00% | 7542 | 0.00% | 7542 | 0.00% | 7542 | 0.00% | 7542 | 0.00% | 7542 | 0.00% | 7542 | 0.00% |
| kroC100 | 20749 | **20749** | **0.00%** | 20749 | 0.00% | 20749 | 0.00% | 20749 | 0.00% | 20749 | 0.00% | 20749 | 0.00% | 20749 | 0.00% | 20749 | 0.00% |
| pr226 | 80369 | **80369** | **0.00%** | 80369 | 0.00% | 80369 | 0.00% | 80369 | 0.00% | 80369 | 0.00% | 80369 | 0.00% | 80369 | 0.00% | 80369 | 0.00% |
| fl417 | 11861 | 11871 | 0.08% | **11862** | **0.01%** | 11867 | 0.05% | 11870 | 0.08% | 11871 | 0.08% | 11863 | 0.02% | 11866 | 0.04% | 11862 | 0.01% |
| kroE100 | 22068 | **22068** | **0.00%** | 22068 | 0.00% | 22068 | 0.00% | 22068 | 0.00% | 22068 | 0.00% | 22068 | 0.00% | 22068 | 0.00% | 22068 | 0.00% |
| pr76 | 108159 | **108159** | **0.00%** | 108159 | 0.00% | 108159 | 0.00% | 108159 | 0.00% | 108159 | 0.00% | 108159 | 0.00% | 108159 | 0.00% | 108159 | 0.00% |
| ch130 | 6110 | 6111 | 0.02% | 6111 | 0.02% | 6111 | 0.02% | 6111 | 0.02% | 6111 | 0.02% | 6111 | 0.02% | **6110** | **0.00%** | 6111 | 0.02% |
| rd100 | 7910 | **7910** | **0.00%** | 7910 | 0.00% | 7910 | 0.00% | 7910 | 0.00% | 7910 | 0.00% | 7910 | 0.00% | 7910 | 0.00% | 7910 | 0.00% |
| tsp225 | 3916 | **3916** | **0.00%** | 3916 | 0.00% | 3916 | 0.00% | 3916 | 0.00% | 3916 | 0.00% | 3916 | 0.00% | 3916 | 0.00% | 3916 | 0.00% |
| pr264 | 49135 | **49135** | **0.00%** | 49135 | 0.00% | 49135 | 0.00% | 49135 | 0.00% | 49135 | 0.00% | 49135 | 0.00% | 49135 | 0.00% | 49135 | 0.00% |
| pr124 | 59030 | **59030** | **0.00%** | 59030 | 0.00% | 59030 | 0.00% | 59030 | 0.00% | 59030 | 0.00% | 59030 | 0.00% | 59030 | 0.00% | 59030 | 0.00% |
| kroA150 | 26524 | **26524** | **0.00%** | 26524 | 0.00% | 26524 | 0.00% | 26524 | 0.00% | 26524 | 0.00% | 26525 | 0.00% | 26524 | 0.00% | 26524 | 0.00% |
| kroB200 | 29437 | **29437** | **0.00%** | 29437 | 0.00% | 29437 | 0.00% | 29437 | 0.00% | 29437 | 0.00% | 29437 | 0.00% | 29437 | 0.00% | 29437 | 0.00% |
| kroB150 | 26130 | **26130** | **0.00%** | 26130 | 0.00% | 26130 | 0.00% | 26130 | 0.00% | 26130 | 0.00% | 26130 | 0.00% | 26130 | 0.00% | 26130 | 0.00% |
| pr107 | 44303 | **44303** | **0.00%** | 44303 | 0.00% | 44303 | 0.00% | 44303 | 0.00% | 44303 | 0.00% | 44303 | 0.00% | 44303 | 0.00% | 44303 | 0.00% |
| lin318 | 42029 | 42080 | 0.12% | **42029** | **0.00%** | 42128 | 0.24% | 42194 | 0.39% | 42029 | 0.00% | 42107 | 0.19% | 42029 | 0.00% | 42029 | 0.00% |
| pr136 | 96772 | **96772** | **0.00%** | 96772 | 0.00% | 96772 | 0.00% | 96772 | 0.00% | 96772 | 0.00% | 96772 | 0.00% | 96772 | 0.00% | 96772 | 0.00% |
| pr299 | 48191 | **48191** | **0.00%** | 48191 | 0.00% | 48191 | 0.00% | 48191 | 0.00% | 48191 | 0.00% | 48191 | 0.00% | 48191 | 0.00% | 48191 | 0.00% |
| u159 | 42080 | **42080** | **0.00%** | 42080 | 0.00% | 42080 | 0.00% | 42080 | 0.00% | 42080 | 0.00% | 42080 | 0.00% | 42080 | 0.00% | 42080 | 0.00% |
| a280 | 2579 | **2579** | **0.00%** | 2579 | 0.00% | 2579 | 0.00% | 2579 | 0.00% | 2579 | 0.00% | 2579 | 0.00% | 2579 | 0.00% | 2579 | 0.00% |
| pr439 | 107217 | 107303 | 0.08% | **107219** | **0.00%** | 107480 | 0.25% | 107810 | 0.55% | 107308 | 0.08% | 107345 | 0.12% | 107228 | 0.01% | 107269 | 0.05% |
| ch150 | 6528 | **6528** | **0.00%** | 6528 | 0.00% | 6528 | 0.00% | 6528 | 0.00% | 6528 | 0.00% | 6528 | 0.00% | 6528 | 0.00% | 6528 | 0.00% |
| d493 | 35002 | 35067 | 0.19% | **35017** | **0.04%** | 35102 | 0.29% | 35151 | 0.43% | 35096 | 0.27% | 35142 | 0.40% | 35039 | 0.11% | 35045 | 0.12% |
| pcb442 | 50778 | 50818 | 0.08% | 50815 | 0.07% | 50810 | 0.06% | 50809 | 0.06% | **50778** | **0.00%** | 50908 | 0.26% | 50807 | 0.06% | 50786 | 0.02% |
| Average | - | 34921 | 0.06% | **34915** | **0.05%** | 34929 | 0.08% | 34941 | 0.10% | 34918 | 0.06% | 34925 | 0.07% | **34915** | **0.05%** | 34916 | 0.05% |

### (b) Medium instances (500–2000 nodes)

| Instance | Optimal | Zero | | Att-GCN | | DIMES | | UTSP | | SoftDist | | DIFUSCO | | Fast-T2T | | GT-Prior | |
|---|---|---|---|---|---|---|---|---|---|---|---|---|---|---|---|---|---|
| | | Length ↓ | Gap ↓ | Length ↓ | Gap ↓ | Length ↓ | Gap ↓ | Length ↓ | Gap ↓ | Length ↓ | Gap ↓ | Length ↓ | Gap ↓ | Length ↓ | Gap ↓ | Length ↓ | Gap ↓ |
| u574 | 36905 | 37150 | 0.66% | 36978 | 0.20% | 37064 | 0.43% | 37088 | 0.50% | 37002 | 0.26% | **36935** | **0.08%** | 36969 | 0.17% | 37001 | 0.26% |
| pcb1173 | 56892 | 57481 | 1.04% | 57283 | 0.69% | 57283 | 0.69% | 57487 | 1.05% | **56968** | **0.13%** | 57084 | 0.34% | 56982 | 0.16% | 57206 | 0.55% |
| u1432 | 152970 | 155871 | 1.90% | 153877 | 0.59% | 153824 | 0.56% | 154276 | 0.85% | 153662 | 0.45% | **153336** | **0.24%** | 153698 | 0.48% | 153542 | 0.37% |
| rat783 | 8806 | 8861 | 0.62% | 8869 | 0.72% | 8865 | 0.67% | 8884 | 0.89% | 8827 | 0.24% | 8820 | 0.16% | 8826 | 0.23% | **8819** | **0.15%** |
| fl1400 | 20127 | 20276 | 0.74% | 20206 | 0.39% | 20234 | 0.53% | 20289 | 0.80% | 20351 | 1.11% | 20253 | 0.63% | 20271 | 0.72% | **20191** | **0.32%** |
| vm1084 | 239297 | 241824 | 1.06% | 239883 | 0.24% | 242129 | 1.18% | 243158 | 1.61% | 240677 | 0.58% | **239492** | **0.08%** | 239841 | 0.23% | 240242 | 0.39% |
| rat575 | 6773 | 6801 | 0.41% | 6807 | 0.50% | 6807 | 0.50% | 6831 | 0.86% | **6780** | **0.10%** | 6783 | 0.15% | 6791 | 0.27% | 6787 | 0.21% |
| vm1748 | 336556 | 340748 | 1.25% | 340010 | 1.03% | 342178 | 1.67% | 343242 | 1.99% | 339937 | 1.00% | **337632** | **0.32%** | 338355 | 0.53% | 339204 | 0.79% |
| rl1889 | 316536 | 321629 | 1.61% | 321778 | 1.66% | 323176 | 2.10% | 325917 | 2.96% | 321654 | 1.62% | **318314** | **0.56%** | 319767 | 1.02% | 321452 | 1.55% |
| u724 | 41910 | 42124 | 0.51% | 42111 | 0.48% | 42128 | 0.52% | 42205 | 0.70% | 42001 | 0.22% | **41982** | **0.17%** | 42031 | 0.29% | 42041 | 0.31% |
| d1291 | 50801 | 51408 | 1.19% | 51208 | 0.80% | 51320 | 1.02% | 51892 | 2.15% | 51220 | 0.82% | **50887** | **0.17%** | 51007 | 0.41% | 51230 | 0.84% |
| pr1002 | 259045 | 261895 | 1.10% | 261505 | 0.95% | 261808 | 1.07% | 261683 | 1.02% | 261075 | 0.78% | 260798 | 0.68% | **260502** | **0.56%** | 260856 | 0.70% |
| fl1577 | 22249 | 22699 | 2.02% | 22531 | 1.27% | 22451 | 0.91% | 22922 | 3.02% | 22686 | 1.96% | 22432 | 0.82% | 22442 | 0.87% | **22350** | **0.45%** |
| nrw1379 | 56638 | 56991 | 0.62% | 56993 | 0.63% | 57013 | 0.66% | 57112 | 0.84% | 57010 | 0.66% | **56787** | **0.26%** | 56968 | 0.58% | 56881 | 0.43% |
| rl1304 | 252948 | 255681 | 1.08% | 254493 | 0.61% | 255372 | 0.96% | 254380 | 0.57% | 254075 | 0.45% | 253518 | 0.23% | **253132** | **0.07%** | 253883 | 0.37% |
| d657 | 48912 | 49107 | 0.40% | 49102 | 0.39% | 49104 | 0.39% | 49151 | 0.49% | 49031 | 0.24% | 48954 | 0.09% | **48943** | **0.06%** | 49034 | 0.25% |
| p654 | 34643 | 34671 | 0.08% | 34663 | 0.06% | 34674 | 0.09% | 34714 | 0.20% | 34757 | 0.33% | 34645 | 0.01% | 34655 | 0.03% | **34643** | **0.00%** |
| d1655 | 62128 | 64249 | 3.41% | 62935 | 1.30% | 63165 | 1.67% | 64179 | 3.30% | 63058 | 1.50% | **62520** | **0.63%** | 62586 | 0.74% | 62789 | 1.01% |
| u1817 | 57201 | 58886 | 2.95% | 58154 | 1.67% | 58677 | 2.58% | 59710 | 4.39% | 58190 | 1.73% | **57842** | **1.12%** | 58083 | 1.54% | 58120 | 1.61% |
| u1060 | 224094 | 227374 | 1.46% | 226136 | 0.91% | 227122 | 1.35% | 229197 | 2.28% | 225843 | 0.78% | 224839 | 0.33% | **224534** | **0.20%** | 224804 | 0.32% |
| rl1323 | 270199 | 272220 | 0.75% | 272431 | 0.83% | 272488 | 0.85% | 275511 | 1.97% | 271577 | 0.51% | **271131** | **0.34%** | 271347 | 0.42% | 271751 | 0.57% |
| Average | - | 123235 | 1.18% | 122759 | 0.76% | 123184 | 0.97% | 123801 | 1.54% | 122684 | 0.74% | **122142** | **0.35%** | 122272 | 0.46% | 122514 | 0.55% |

### (c) Large instances (2000+ nodes)

| Instance | Optimal | Zero | | Att-GCN | | DIMES | | UTSP | | SoftDist | | DIFUSCO | | Fast-T2T | | GT-Prior | |
|---|---|---|---|---|---|---|---|---|---|---|---|---|---|---|---|---|---|
| | | Length ↓ | Gap ↓ | Length ↓ | Gap ↓ | Length ↓ | Gap ↓ | Length ↓ | Gap ↓ | Length ↓ | Gap ↓ | Length ↓ | Gap ↓ | Length ↓ | Gap ↓ | Length ↓ | Gap ↓ |
| u2152 | 64253 | 67471 | 5.01% | 66068 | 2.82% | 66843 | 4.03% | 68491 | 6.60% | 65839 | 2.47% | **65516** | **1.97%** | 66106 | 2.88% | 65551 | 2.02% |
| u2319 | 234256 | 240322 | 2.59% | 235175 | 0.39% | 235562 | 0.56% | 236094 | 0.78% | **234601** | **0.15%** | 235421 | 0.50% | 235135 | 0.38% | 234929 | 0.29% |
| pcb3038 | 137694 | 143874 | 4.49% | 140016 | 1.69% | 142489 | 3.48% | 141517 | 2.78% | 138962 | 0.92% | 141604 | 2.84% | 140908 | 2.33% | **138911** | **0.88%** |
| fl3795 | 28772 | 32286 | 12.21% | 30944 | 7.55% | 31063 | 7.96% | 31397 | 9.12% | 30140 | 4.75% | 30880 | 7.33% | **29907** | **3.94%** | 30546 | 6.17% |
| pr2392 | 378032 | 386207 | 2.16% | 386482 | 2.24% | 386655 | 2.28% | 383247 | 1.38% | 383037 | 1.32% | 387877 | 2.60% | **381546** | **0.93%** | 381598 | 0.94% |
| fnl4461 | 182566 | 188373 | 3.18% | 185826 | 1.79% | 186745 | 2.29% | 188081 | 3.02% | **184360** | **0.98%** | 186633 | 2.23% | 187670 | 2.80% | 184471 | 1.04% |
| d2103 | 80450 | 83672 | 4.00% | 82438 | 2.47% | 83990 | 4.40% | 84567 | 5.12% | 82022 | 1.95% | 81886 | 1.78% | 82387 | 2.41% | **81010** | **0.70%** |
| rl5934 | 556045 | 595415 | 7.08% | 577629 | 3.88% | 590654 | 6.22% | 616033 | 10.79% | **572562** | **2.97%** | 579799 | 4.27% | 576954 | 3.76% | 574982 | 3.41% |
| rl5915 | 565530 | 594801 | 5.18% | 588374 | 4.04% | 592293 | 4.73% | 604537 | 6.90% | **581947** | **2.90%** | 584322 | 3.32% | 588056 | 3.98% | 584110 | 3.29% |
| usa13509 | 19982859 | 21193040 | 6.06% | 20972608 | 4.95% | 21240984 | 6.30% | 21425850 | 7.22% | 21329926 | 6.74% | 21165172 | 5.92% | 20919478 | 4.69% | **20642852** | **3.30%** |
| brd14051 | 469385 | 495040 | 5.47% | 488704 | 4.12% | 492982 | 5.03% | 498371 | 6.18% | 489854 | 4.36% | 487329 | 3.82% | 487370 | 3.83% | **482231** | **2.74%** |
| d18512 | 645238 | 679290 | 5.28% | 672809 | 4.27% | 675878 | 4.75% | 683918 | 5.99% | 671251 | 4.03% | 669281 | 3.73% | 670876 | 3.97% | **662290** | **2.64%** |
| rl11849 | 923288 | 990945 | 7.33% | 974291 | 5.52% | 987813 | 6.99% | 1002904 | 8.62% | 970110 | 5.07% | 962745 | 4.27% | 971991 | 5.27% | **959355** | **3.91%** |
| d15112 | 1573084 | 1657644 | 5.38% | 1641612 | 4.36% | 1646903 | 4.69% | 1668137 | 6.04% | 1633396 | 3.83% | 1636603 | 4.04% | 1636986 | 4.06% | **1614104** | **2.61%** |
| Average | - | 1953455 | 5.39% | 1931641 | 3.58% | 1954346 | 4.55% | 1973796 | 5.75% | 1954857 | 3.03% | 1943933 | 3.47% | 1926812 | 3.23% | **1902638** | **2.42%** |

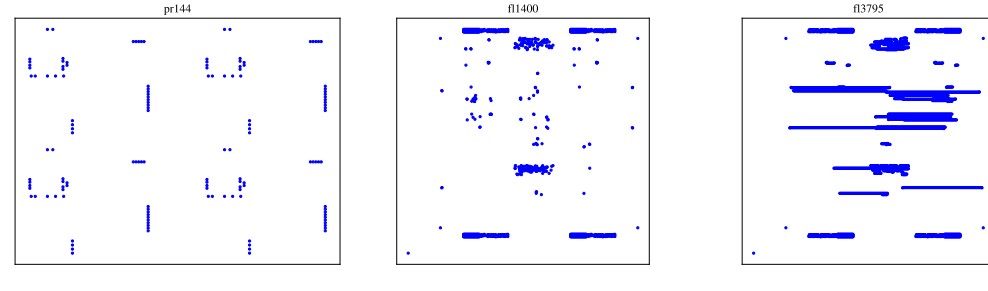

(a) Hard instances at small, medium, and large scales.

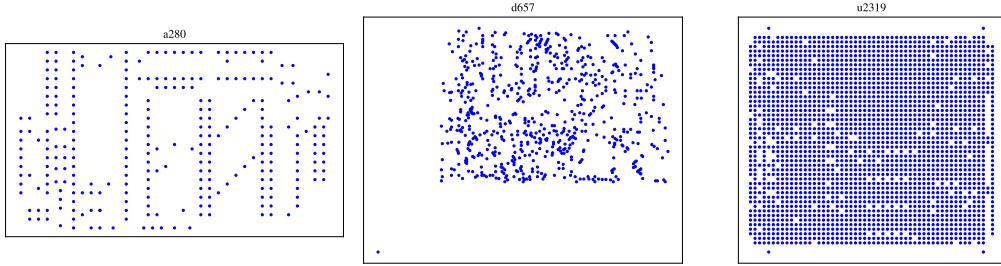

(b) Easy instances at small, medium, and large scales.

Figure 6: Representative TSPLIB instances visualization.

Table 12: The MCTS hyperparameter search space. Bolded configurations indicate default settings from prior works.

| Hyperparameter | Range |
|---|---|
| Alpha | $[0, \mathbf{1}, 2]$ |
| Beta | $[\mathbf{10}, 100, 150]$ |
| Max_Depth | $[\mathbf{10}, 50, 100, 200]$ |
| Max_Candidate_Num | $[5, 20, 50, \mathbf{1000}]$ |
| Param_H | $[2, 5, \mathbf{10}]$ |
| Use_Heatmap | $[\mathbf{True}, False]$ |

# I HYPERPARAMETER TUNING WITH SMAC3

In addition to the grid search method employed in the main content of this paper, we also conducted hyperparameter tuning using the Sequential Model-based Algorithm Configuration (SMAC3) framework (Lindauer et al., 2022). SMAC3 is designed for optimizing algorithm configurations through an efficient and adaptive search process that balances exploration and exploitation of the hyperparameter space.

The SMAC3 framework utilizes a surrogate model to guide sequential model-based (Bayesian) hyperparameter optimization. This model is iteratively refined as configurations are evaluated, allowing SMAC3 to identify promising areas of the search space more effectively than traditional methods. This provides a principled and reproducible way to select MCTS settings under a fixed evaluation budget.

Table 13: The Comparison of Tuning Time Between Grid Search and SMAC3. "h" indicates hours.

| | Grid Search | SMAC3 |
|---|---|---|
| TSP-500 | 24h | 1.39h |
| TSP-1000 | 48h | 2.78h |
| TSP-10000 | 6h | 3.47h |

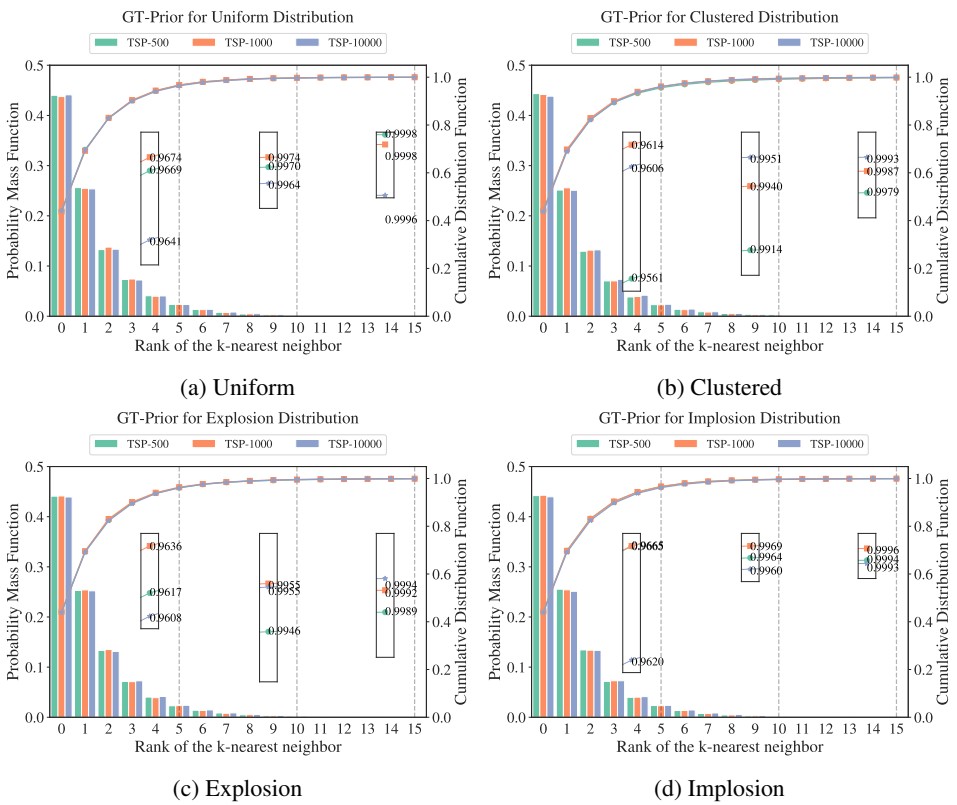

Figure 7: Empirical distribution of $k$-nearest neighbor in optimal TSP tours of different distributions.

For our experiments, we configured SMAC3 to optimize the same hyperparameters as those previously tuned via grid search. The search space remains identical to that demonstrated in Table 12. However, we set SMAC3 to search for 50 epochs (50 different hyperparameter combinations) instead of exploring the entire search space (864 different combinations) and the time limit for MCTS was set to 50 seconds for TSP-500, 100 seconds for TSP-1000, and 1000 seconds for TSP-10000. We show the time cost of each tuning method in Table 13.

The results of these experiments, including the hyperparameter settings identified by SMAC3 and their corresponding performance metrics, are presented in Tables 14 and 15. As shown, the performance achieved by SMAC3 is comparable to that of grid search. Specifically, for TSP-500 and TSP-1000, SMAC3 produces results similar to those of Att-GCN DIFUSCO and GT-Prior, with even better outcomes observed on TSP-10000. This improvement can be attributed to the extended tuning time allowed by SMAC3 compared to grid search. Given the significant difference in time costs, SMAC3 proves to be an efficient and economical option for tuning MCTS hyperparameters.

## J $k$-NEAREST NEIGHBOR PRIOR IN TSP INSTANCES WITH DIFFERENT DISTRIBUTIONS

We solved and analyzed TSP problem instances across several different distributions and found that their $k$-Nearest Neighbor Prior distribution similarities were quite high, as shown in the Figure 7.

## K ABLATION STUDY ON THE EFFICACY OF HYPERPARAMETER TUNING

To better understand the efficacy of hyperparameter tuning in MCTS for solving TSP, we conducted an ablation study focusing on two critical aspects: the relationship between search time and solution quality, and the sample efficiency of our tuning process. These experiments provide valuable insights into our algorithm's performance characteristics and highlight areas for potential optimization.

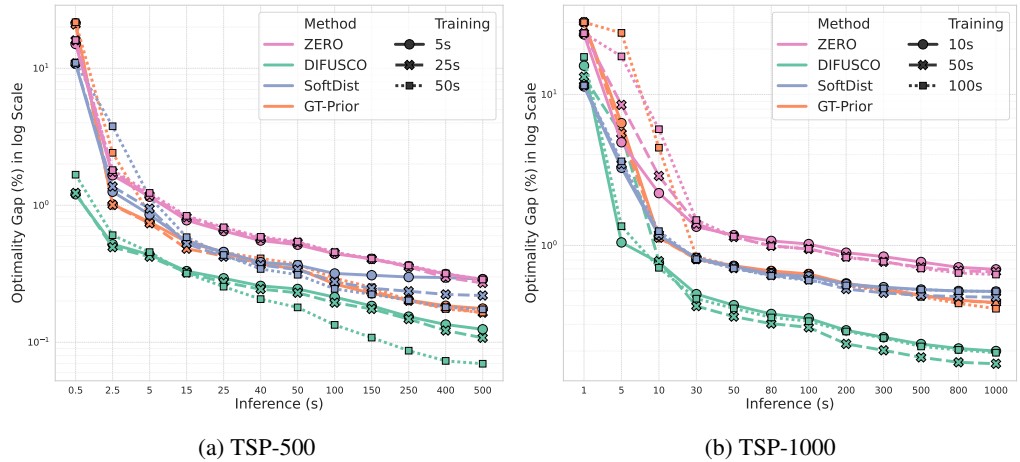

(a) TSP-500              (b) TSP-1000

Figure 8: Impact of search time on solver performance across different hyperparameter configurations.

### K.1 IMPACT OF TUNING STAGE TIME_LIMIT ON SOLVER PERFORMANCE

The relationship between `Time_Limit` and hyperparameter quality is crucial in MCTS hyperparameter tuning. While longer search times might intuitively yield better results, they also lead to significantly increased tuning time. We conducted an ablation study to investigate this trade-off and seek a balance between performance and efficiency.

**Experimental Setup** We examined the impact of search time on solver performance for TSP-500 and TSP-1000 instances, varying the tuning stage `Time_Limit` from 0.1 to 0.05 and 0.01.

Figure 8 shows the performance of different methods with varying inference times, each with three hyperparameter sets tuned using different `Time_Limit` values. Surprisingly, the relative performance remains largely consistent across search durations, suggesting that hyperparameter effectiveness can be accurately assessed within a limited time frame.

For TSP-500, most heatmaps exhibit similar performance across all tuning stage `Time_Limit` values, with Zero and GT-Prior methods showing nearly identical performance curves. The best learning-based method, DIFUSCO, displays a small performance gap at the default 50-second inference time limit. However, this gap widens with longer inference times, suggesting that optimal MCTS settings for high-quality heatmaps may vary with different `Time_Limit` values during tuning phase. Efficiently tuning hyperparameters for such high-quality heatmaps remains a future research direction. Notably, TSP-1000 results show even smaller performance gaps between different tuning stage `Time_Limit` values, indicating that shorter tuning times can yield satisfactory hyperparameter settings for larger problem instances.

The consistency of relative performance across search times has significant implications for efficient hyperparameter tuning in large-scale TSP solving. This insight enables the development of accelerated evaluation procedures that can identify promising hyperparameter settings without exhaustive, long-duration searches.

### K.2 SAMPLE EFFICIENCY

Experiments were conducted to evaluate the sample efficiency of the hyperparameter tuning procedure for our proposed $k$-nearest prior heatmap. By varying the number of TSP instances in the training set and measuring the resulting solution quality of the tuned hyperparameter setting, insights were gained into the computational efficiency of our method. With only 64 samples for hyperparameter tuning, our proposed GT-Prior achieved a gap of 0.493% on TSP-500 and 0.866% on TSP-1000, rivaling the performance of hyperparameter tuning with 256 samples, which achieved 0.493% on TSP-500 and 0.858% on TSP-1000. These results demonstrate the high sample efficiency of our approach, enabling effective tuning with minimal computational resources.

## L GT-PRIOR INFORMATION

Table 14: Tuned hyperparameters of all the methods for TSP-500, TSP-1000 and TSP-10000 by grid search (the left table) and SMAC3 (right). UH: 0=`False`, 1=`True`.

| | METHOD | ALPHA | BETA | H | MCN | UH | MD | | METHOD | ALPHA | BETA | H | MCN | UH | MD |
|---|---|---|---|---|---|---|---|---|---|---|---|---|---|---|---|
| | ZERO | 2 | 10 | 2 | 5 | 0 | 100 | | ZERO | 0 | 150 | 2 | 5 | 0 | 50 |
| | ATT-GCN | 0 | 150 | 5 | 5 | 0 | 100 | | ATT-GCN | 2 | 150 | 2 | 5 | 0 | 100 |
| | DIMES | 0 | 100 | 5 | 5 | 0 | 200 | | DIMES | 0 | 100 | 5 | 5 | 0 | 200 |
| | DIFUSCO | 1 | 150 | 2 | 5 | 0 | 50 | | DIFUSCO | 1 | 150 | 2 | 5 | 0 | 50 |
| TSP-500 | UTSP | 0 | 100 | 5 | 5 | 0 | 50 | TSP-500 | UTSP | 0 | 100 | 5 | 5 | 0 | 50 |
| | SOFTDIST | 1 | 100 | 5 | 20 | 0 | 200 | | SOFTDIST | 1 | 100 | 5 | 20 | 0 | 200 |
| | FAST-T2T | 1 | 150 | 2 | 20 | 0 | 200 | | FAST-T2T | 1 | 10 | 2 | 20 | 0 | 50 |
| | GT-PRIOR | 0 | 10 | 5 | 5 | 1 | 200 | | GT-PRIOR | 0 | 10 | 5 | 5 | 1 | 200 |
| | ZERO | 1 | 100 | 5 | 5 | 0 | 100 | | ZERO | 0 | 150 | 2 | 5 | 0 | 100 |
| | ATT-GCN | 0 | 150 | 5 | 5 | 0 | 200 | | ATT-GCN | 2 | 150 | 2 | 5 | 0 | 100 |
| | DIMES | 0 | 150 | 2 | 5 | 0 | 200 | | DIMES | 0 | 150 | 5 | 5 | 0 | 100 |
| | DIFUSCO | 0 | 150 | 2 | 5 | 1 | 100 | | DIFUSCO | 0 | 150 | 2 | 5 | 1 | 200 |
| TSP-1000 | UTSP | 1 | 100 | 5 | 5 | 0 | 50 | TSP-1000 | UTSP | 0 | 100 | 5 | 5 | 0 | 50 |
| | SOFTDIST | 0 | 150 | 2 | 20 | 1 | 200 | | SOFTDIST | 1 | 100 | 2 | 50 | 1 | 200 |
| | FAST-T2T | 0 | 150 | 2 | 1000 | 1 | 50 | | FAST-T2T | 1 | 10 | 2 | 50 | 1 | 50 |
| | GT-PRIOR | 1 | 10 | 5 | 5 | 1 | 200 | | GT-PRIOR | 0 | 150 | 2 | 5 | 1 | 200 |
| | ZERO | 0 | 100 | 2 | 20 | 0 | 10 | | ZERO | 0 | 100 | 2 | 20 | 0 | 10 |
| | ATT-GCN | 1 | 150 | 2 | 5 | 1 | 50 | | ATT-GCN | 1 | 150 | 2 | 5 | 1 | 50 |
| | DIMES | 1 | 100 | 2 | 20 | 0 | 10 | | DIMES | 1 | 100 | 2 | 20 | 0 | 10 |
| TSP-10000 | DIFUSCO | 0 | 100 | 5 | 20 | 0 | 50 | TSP-10000 | DIFUSCO | 0 | 100 | 5 | 20 | 0 | 50 |
| | SOFTDIST | 2 | 100 | 5 | 20 | 0 | 10 | | SOFTDIST | 2 | 100 | 5 | 20 | 0 | 10 |
| | FAST-T2T | 2 | 150 | 2 | 20 | 0 | 10 | | FAST-T2T | 1 | 10 | 10 | 50 | 0 | 200 |
| | GT-PRIOR | 1 | 100 | 10 | 1000 | 1 | 100 | | GT-PRIOR | 1 | 100 | 10 | 1000 | 1 | 100 |

Table 15: Results of Hyperparameter Tuning using SMAC3. The underlined figures in the table indicate results that are equal to or better than those of Grid Search, rounded to two decimal places.

| METHOD | TYPE | TSP-500 | | | TSP-1000 | | | TSP-10000 | | |
|---|---|---|---|---|---|---|---|---|---|---|
| | | LENGTH ↓ | GAP ↓ | TIME ↓ | LENGTH ↓ | GAP ↓ | TIME ↓ | LENGTH ↓ | GAP ↓ | TIME ↓ |
| CONCORDE | OR(EXACT) | 16.55* | — | 17.65s | 23.12* | — | 3.12M | N/A | N/A | N/A |
| GUROBI | OR(EXACT) | 16.55 | 0.00% | 21.39M | N/A | N/A | N/A | N/A | N/A | N/A |
| LKH-3 (DEFAULT) | OR(HEURISTIC) | 16.55 | 0.00% | 14.84s | 23.12 | 0.00% | 1.02M | 71.77* | — | 28.73M |
| ZERO | MCTS | 16.67 | 0.73% | 0.00M+ 50.06s | 23.39 | 1.17% | 0.00M+ 1.67M | 74.44 | 3.71% | 0.00M+ 16.65M |
| ATT-GCN† | SL+MCTS | 16.66 | 0.69% | 0.52M+ 50.06s | 23.38 | 1.15% | 0.73M+ 1.67M | 73.87 | 2.92% | 4.16M+ 16.65M |
| DIMES† | RL+MCTS | 16.67 | 0.73% | 0.97M+ 50.06s | 23.42 | 1.31% | 2.08M+ 1.67M | 74.17 | 3.33% | 4.65M+ 16.65M |
| UTSP† | UL+MCTS | 16.72 | 1.07% | 1.37M+ 50.06s | 23.51 | 1.68% | 3.35M+ 1.67M | — | — | — |
| SOFTDIST† | SOFTDIST+MCTS | 16.62 | 0.46% | 0.00M+ 50.06s | 23.33 | 0.90% | 0.00M+ 1.67M | 75.34 | 4.97% | 0.00M+ 16.65M |
| DIFUSCO† | SL+MCTS | 16.62 | 0.43% | 3.61M+ 50.06s | **23.24** | **0.53%** | 11.86M+ 1.67M | **73.26** | **2.06%** | 28.51M+ 16.65M |
| FAST-T2T | SL+MCTS | **16.60** | **0.34%** | 0.50M+ 50.06s | 23.29 | 0.74% | 1.78M+ 1.67M | 75.11 | 4.65% | 7.73M+ 16.65M |
| GT-PRIOR | PRIOR+MCTS | 16.63 | 0.50% | 0.00M+ 50.06s | 23.32 | 0.85% | 0.00M+ 1.67M | 73.26 | 2.07% | 0.00M+ 16.65M |

We provide detailed information about GT-Prior for constructing the heatmap for TSP-500, TSP-1000, and TSP-10000 as follows:

**GT-PRIOR raw probabilities.** `TSP-500`:
```
4.40078125e-01,2.56265625e-01,1.32750000e-01,7.32656250e-02,
4.08125000e-02,2.35937500e-02,1.34062500e-02,7.75000000e-03,
4.48437500e-03,2.73437500e-03,1.78125000e-03,1.18750000e-03,
6.87500000e-04,3.75000000e-04,3.75000000e-04,1.87500000e-04,
7.81250000e-05,1.56250000e-05,4.68750000e-05,1.56250000e-05,
4.68750000e-05,3.12500000e-05,1.56250000e-05,1.56250000e-05.
```

`TSP-1000`:
```
4.37554687e-01,2.54718750e-01,1.37671875e-01,7.41093750e-02,
3.97890625e-02,2.35156250e-02,1.32265625e-02,7.45312500e-03,
4.73437500e-03,3.00781250e-03,1.59375000e-03,1.08593750e-03,
5.62500000e-04,2.96875000e-04,2.65625000e-04,1.71875000e-04,
1.01562500e-04,4.68750000e-05,1.56250000e-05,3.12500000e-05,
2.34375000e-05,7.81250000e-06,1.56250000e-05.
```

```
TSP-10000:
4.4175625e-01,2.5409375e-01,1.3292500e-01,7.1950000e-02,3.9518750e-02,
2.3750000e-02,1.4143750e-02,8.0937500e-03,4.9125000e-03,3.3312500e-03,
1.8437500e-03,1.1125000e-03,8.3750000e-04,5.5625000e-04,3.7500000e-04,
2.6250000e-04,1.8125000e-04,8.7500000e-05,6.8750000e-05,5.0000000e-05,
5.0000000e-05,2.5000000e-05,2.5000000e-05,6.2500000e-06,1.2500000e-05,
6.2500000e-06,6.2500000e-06,6.2500000e-06,6.2500000e-06,6.2500000e-06.
```

## M  LIMITATIONS AND FUTURE WORK

Our study, while highlighting the critical role of MCTS configuration and the efficacy of simple priors, has several limitations that suggest avenues for future research:

- **Scope of TSP Variants and MCTS Adaptation:** The current analysis, including the GT-Prior, focuses on Euclidean TSP, and the MCTS framework utilizes TSP-specific $k$-opt moves. The direct applicability of our findings and the GT-Prior to non-Euclidean TSPs, other combinatorial optimization problems, or different MCTS action spaces warrants further investigation.

- **Empirical Nature of MCTS Tuning:** While we demonstrate the profound impact of MCTS tuning, our approach to finding optimal configurations is empirical. A deeper theoretical understanding of the relationship between TSP instance properties (or heatmap characteristics) and optimal MCTS hyperparameters could lead to more principled, instance-adaptive tuning strategies, reducing the reliance on extensive offline searches.

- **Hyperparameter Tuning Efficiency:** While the proposed GT-Prior heatmap is computationally inexpensive at inference, the MCTS hyperparameter tuning process itself (using grid search in our current implementation) can be resource-intensive, especially if the search space is large or if tuning is performed on very large-scale instances. Although this is a one-time offline cost, optimizing the tuning process itself (e.g., using more sophisticated Bayesian optimization, evolutionary algorithms, or meta-learning for hyperparameter optimization as hinted in Appendix I) would be beneficial for practical adoption and for exploring even larger parameter spaces.

- **Exploration of Alternative Search Mechanisms:** This study operates within the MCTS framework as the search component. While MCTS is powerful, exploring whether the insights on the heatmap vs. search balance extend to other search metaheuristics (e.g., guided local search, iterated local search, or even learned search policies) when paired with various heatmap generation techniques could be a valuable research direction.

Addressing these aspects could lead to more versatile, theoretically grounded, and practically efficient learning-based solvers for TSP and other challenging optimization problems.

## N  LLM USAGE STATEMENT

During the preparation of this manuscript, we utilized a large language model (LLM) as a writing assistant. The LLM's role was strictly limited to improving the clarity, grammar, and readability of our text through sentence polishing and paragraph restructuring. The LLM did not contribute to research ideation, experimental design, data analysis, or the formulation of conclusions. All scientific content and claims are the sole responsibility of the human authors.

