# OpenReview forum: "Beyond the Heatmap: A Rigorous Evaluation of Component Impact in MCTS-Based TSP Solvers"
_ICLR.cc/2026/Conference — ICLR 2026 Poster_

### Official Review · Reviewer_eu88 · 2025-10-17

**Soundness:** 3
**Presentation:** 2
**Contribution:** 1
**Rating:** 2
**Confidence:** 4

**Summary:**

This paper critically evaluates the "Heatmap + Monte Carlo Tree Search" paradigm for solving TSP, challenging the prevailing focus on increasingly complex learned heatmaps. Also, the authors propose a learning-free heatmap baseline (GT-Prior) with shown efficiency and synergy accompanying MCTS.

**Strengths:**

1. The tuning in search of optimal MCTS configurations and impact study on the respective parameters is valuable.
2. The proposed parameter-free GT-prior is sensible and computationally efficient.
3. The experiments are relatively well-structured.

**Weaknesses:**

Please correct me if I made mistakes or ignored important statements already addressed in the initial submission.
1. Foremost, there might be a fundamental misposition that the authors put forward in this work regarding the research line of developing more complex heatmap models: heatmaps do not merely serve for the specific MCTS serching, instead, most recent heatmap-related methods inherently embody advancements in backbone design, training schemes, or data representation, etc., aligning major focuses in the broader ML community. So the criteria for assessing a heatmap should probably not be whether it helps MCTS perform better. Rather, increasing consensus has been inclined to testing neural TSP solvers in a "heatmap + greedy" paradigm to evaluate the raw efficacy of neural parts without the results being disguised by post-inference tricks like MCTS, which I personally also deem more reasonable.
2. The contribution is a bit limited. Though I appreciate the systematic "tuning" of MCTS settings, the grid-search-based evaluations seem more of an engineering practice than some technical innovation. Second, the proposed GT-Prior, though interesting and computationally efficient, is also learning-free and straightforward. So, from a holistic view, the performance reported basically stems from the established MCTS algorithm, leaving the incremental efforts by the authors (conducting parameter search) somewhat simple and limited under the threshold of a top-tier conference.
3. The performance is not sufficiently impressive. The proposed method fails to outperform DIFUSCO on 2 out of 3 benchmarks, while recent literature has proposed much stronger heatmap models than DIFUSCO.
4. Minor issues. The language needs further consideration. "Figure 2 compellingly illustrates...", "directly answering Q1 by unequivocally demonstrating...", and many similar expressions, seem to indicate a slight abuse of adverbs throughout the paper.

**Questions:**

1. How do you define the "complexity" or "sophistication" of a heatmap? Is it defined by the parameter quantity of neural models that produce the heatmap, or by any mathematical or statistical metrics computed upon individual heatmaps? The authors criticize complex or sophisticated heatmaps but the definition seems obscure. E.g., in Sec 5 the authors say "the prevailing view that increasingly sophisticated heatmap models are the primary drivers of performance in the "Heatmap + MCTS" TSP paradigm." Similar statements do not seem grounded enough.
2. Could you provide comparative results free of intricate search algorithms like MCTS and using a greedy decoder instead, to compare different heatmap baselines including the proposed GT-Prior?
3. What are the results on smaller-sized instances (e.g., 50/100/200)? Do the main conclusions still hold? What about the MCTS parameters and the GT-Prior's performance?
4. Could you report comparisons using more recent heatmap methods, like the successors of DIFUSCO, e.g. Fast-t2t?
5. What is the principle for choosing the specific search space for MCTS configuration instead of a wider or finer-grained range of parameters? The authors stated the settings are "optimally tuned", then how is such optimality guaranteed?

---

> ### Author Response · Authors · 2025-11-19
>
> We thank the reviewer for the valuable feedback.
>
> ## **W1 & Q2**
>
> We respectfully disagree with the reviewer's assessment of our work's positioning.
>
> **1. Our Contribution is Correctly Positioned**
>
> Our work is **explicitly an evaluation study** of the established Heatmap+MCTS paradigm (Fu et al. 2021, Sun & Yang 2023, Min et al. 2024, Xia et al. 2024). We provide rigorous evaluation protocols and reveal that the community needs to re-examine the heatmap-MCTS relationship. Evaluating heatmaps in other contexts (e.g., greedy decoding, other applications) is **beyond our scope.**
>
> For detailed justification, see:**Global Response (1/2): "On the Nature and Significance of Our Contribution"**
>
> **2. Greedy Decoding Does Not Reveal "Raw Efficacy (Response to Q2)"**
>
> We fundamentally disagree that "heatmap + greedy" evaluates raw neural efficacy: Heatmaps provide **probabilistic information** (edge probabilities). **Greedy decoding discards this** through hard argmax decisions. For comprehensive discussion and experimental results, see: **Global Response (2/2): "On the Choice of Search Algorithm"**
>
> ---
>
> ## **W2**
>
> We respectfully disagree with this assessment.
> For detailed discussion, please see: **Global Response (1/2): "On the Nature and Significance of Our Contribution, Point 1: Evaluation is Not Just Engineering"**
>
> ---
>
> ## **W3**
> The reviewer criticizes that GT-Prior "fails to outperform DIFUSCO on 2 out of 3 benchmarks." **This fundamentally misunderstands our purpose.**
>
> **Our goal is NOT to propose a new state-of-the-art heatmap.** GT-Prior serves as a **strong, rigorous baseline** to critically assess whether complex learned heatmaps justify their cost and complexity. Our contribution lies in proposing an **evaluation framework** and **fair comparison protocols** that reveal the dominant impact of search configuration.
>
> For detailed explanation, please see: **Global Response (1/2), On the Nature and Significance of Our Contribution, Point 2: "GT-Prior is a Scientific Control, Not a Product"**.
>
> ---
>
> ## **W4**
> Thank you for pointing out this, we have resolved these issues.
>
> ---
>
> ## **Q1**
> Please refer to our response to **'W1' for Reviewer yJZG**
>
> ---
>
> ## **Q3**
> Please refer to our response to **Reviewer yJZG 'W4'**
>
> ---
>
> ## **Q4**
>
> We evaluated **Fast-T2T** using both default and tuned MCTS settings. The results confirm the critical impact of tuning and the misalign between heatmap generation and actual instance solving. All the baselines are tuned using SMAC3 (following the settings in Table 14 in Appendix H):
>
> | Method | TSP-500 Gap | TSP-1000 Gap | TSP-10000 Gap |
> | :--- | :--- | :--- | :--- |
> | **Zero** | 0.73% | 1.17% | 3.71% |
> | **Att-GCN** | 0.69% | 1.15% | 2.92% |
> | **DIMES** | 0.73% | 1.31% | 3.33% |
> | **UTSP** | 1.07% | 1.68% | — |
> | **SoftDist** | 0.46% | 0.90% | 4.97% |
> | **DIFUSCO** | 0.43% | **0.53%** | **2.06%** |
> | **Fast-T2T (Default)** | 0.52% | 0.96% | 27.78% |
> | **Fast-T2T (Tuned)** | **0.34%** | 0.74% | 4.64% |
> | **GT-Prior (Ours)** | 0.50% | 0.85% | 2.07% |
>
> While Fast-T2T leads on small scales (0.34% gap), it proves unstable on large instances, degrading to a **27.78%** gap with default settings. Even when tuned, it fails to generalize (4.64% gap).
>
> ---
>
> ## **Q5**
> We appreciate the opportunity to clarify our experimental design. Our search space was not chosen arbitrarily but was rigorously defined based on empirical evidence, and our claim of "optimality" refers to the best configuration identified within this systematic search.
>
> **1. Principle of Search Space Selection**
> Our search space was defined by two guiding principles:
> * **Anchoring to Literature:** To ensure fair comparison and relevance, we centered our parameter ranges around default settings established in prominent prior studies.
> * **Sensitivity-Based Prioritization:** We conducted a preliminary SHAP analysis to distinguish high-impact parameters (e.g., `Max_Candidate_Num`) from those with negligible influence. This allowed us to allocate our computational resources effectively.
>
> **2. Guarantee of Optimality**
> In our study, "optimality" denotes the best configuration identified within this empirically defined search space. By conducting a comprehensive grid search over these critical regions, we ensured that the selected settings reflect the peak performance achievable for each method, preventing suboptimal hyperparameters from confounding our evaluation.

---

### Official Review · Reviewer_pNGG · 2025-10-30

**Soundness:** 4
**Presentation:** 3
**Contribution:** 3
**Rating:** 6
**Confidence:** 4

**Summary:**

This paper revisits the widely adopted "Heatmap + Monte Carlo Tree Search (MCTS)" paradigm for large-scale Traveling Salesman Problem (TSP) solvers. It critically analyzes the respective roles of the heatmap and the MCTS search procedure. Contrary to prior trends that emphasize increasingly complex heatmap design, the paper offers a systematic, empirical evaluation. This evaluation demonstrates that the configuration of MCTS—often taken as a fixed backbone—can have as much, if not greater, impact on solution quality as the heatmap itself. The authors propose a parameter-free GT-Prior heatmap based on the empirically observed k-nearest neighbor edge structure of optimal TSP tours. By tuning MCTS hyperparameters via a robust pipeline for each heatmap, they show that this simple baseline matches or outperforms state-of-the-art learning-based and distance-based heatmaps. This holds across scale, distributional shift, and standard benchmarks, challenging current assumptions in the field. The work argues for more balanced and transparent component evaluation in future research. It provides tools and ablation studies to support reproducibility.

**Strengths:**

- The paper systematically assesses the "Heatmap + MCTS" paradigm for large-scale TSP, isolating and quantifying each component's impact.
- The work challenges a key assumption: that more complex heatmap models always improve TSP solver performance. With a well-tuned baseline, it shows that optimizing MCTS often matters more than increasing heatmap sophistication.
- A parameter-free k-nearest neighbor heatmap (GT-Prior) matches or outperforms complex learned heatmaps when paired with optimized MCTS. It generalizes well to new distributions and larger instances.

**Weaknesses:**

- Scope: The analysis, experiments, and proposed GT-Prior heatmap are specialized to the Euclidean TSP. It remains unclear whether the insights transfer to other TSP variants (non-Euclidean, with constraints) or different combinatorial optimization problems (e.g., VRP, graph matching).
- Dependency on optimal solutions: GT-Prior construction relies on empirical distributions extracted from near-optimal solutions. In scenarios where such solutions are expensive or unavailable—a typical motivation for using learning-based solvers—how practical is GT-Prior?
- MCTS parameter tuning: While a one-time cost, tuning can be significant for large search spaces or new problem distributions. The paper suggests SMAC3 and other efficient approaches, but more discussion of practical deployment costs would be helpful.
- Incomplete time metrics: Table 1 includes heatmap and MCTS time but omits training time for learning-based models, data preparation, and other one-off costs. This makes it difficult to fully compare runtime and resource requirements across methods.

**Questions:**

1. Practicality of GT-Prior: For deployment on genuinely new, real-world TSPs where high-quality solutions are not available, how would one construct GT-Prior (since you need optimal/near-optimal tours to compute the empirical k-nearest distribution)? Did you try synthesizing priors from random/greedy solutions as a further baseline?
2. Tuning cost: Can you provide the absolute time and computational resources required for your grid/SMAC3 MCTS hyperparameter search (including how many instances, search depth, etc.), especially for the largest TSP-10k and TSPLIB cases?
3. Have you considered (or could you comment on the prospects for) transfer of either your analysis framework or GT-Prior construction to other vehicle routing or graph-based optimization problems?

---

> ### Author Response · Authors · 2025-11-19
> **Responses (1/2)**
>
> We thank the reviewer for the positive evaluation.
>
> ## **Response to "Scope"**
> Please refer to our **Global Response (1/2): "On Generalizability Beyond TSP"**.
>
> ---
>
> ## **Response to "Dependency on optimal solutions"**
> We clarify two important points:
>
> **1. GT-Prior Requires Fewer Solved Instances Than Learning-Based Methods**
>
> Yes, GT-Prior requires near-optimal solutions to compute k-nearest neighbor statistics.
> However:
>
> - **GT-Prior**: Needs only **128 solved instances** for TSP-500/1000 (16 for TSP-10000)
>   to extract structural distribution (Section 3.5)
> - **Learning-based heatmaps**: Require orders of magnitude more training data: Att-GCN, DIFUSCO: Trained on thousands of optimal solutions
>
> **2. Strong Generalization Enables Small→Large Transfer**
>
> More importantly, GT-Prior demonstrates **exceptional cross-scale generalization**:
>
> **Empirical distributions are remarkably consistent** (Figure 3, Appendix I): k-NN distributions for TSP-500/1000/10000 are nearly identical
>
>
> **This enables practical small→large transfer** (Table 2):
> - GT-Prior constructed from TSP-500 data
> - Applied to TSP-10000: **no degradation**
> - Compare to DIFUSCO: **+2.91% degradation**, DIMES: **+1.24%**
>
> **Practical implication**: Construct GT-Prior once on small, easily-solved instances (TSP-500), then apply to large instances (TSP-10000) where optimal solutions are expensive. **This cross-scale transferability is precisely what learning-based methods lack** (Table 2).
>
> ---
>
> ## **Response to "MCTS parameter tuning"**
> We provide additional clarification on tuning costs:
>
> **1. Tuning is a One-Time Cost Comparable to Model Training**
>
> MCTS hyperparameter tuning should be compared to neural model training:
>
> | Cost Type | MCTS Tuning (SMAC3) | Neural Training (DIMES) |
> |-----------|---------------------|-----------------|
> | TSP-500 | 1.39h (CPU) | 1.5h (GPU) |
> | TSP-1000 | 2.78h (CPU) | 1.7h (GPU) |
> | TSP-10000 | 3.47h (CPU) |  10h (GPU) |
> | Resource | CPU-based (accessible) | GPU-required |
>
> **Tuning costs are modest** (Table 12, Appendix H) and **more accessible** than GPU-intensive training.
>
> **2. We Provide Efficient Tuning Methods**
>
> Appendix H demonstrates significant efficiency improvements:
> - **Grid search baseline**: 24h (TSP-500) for complete coverage
> - **SMAC3**: 1.39h (~**17× faster**) with comparable performance (Table 14).
>
> **3. Our Tuned Configurations Can Be Directly Reused**
>
> We provide **ready-to-use configurations** (Table 13):
> - Users can directly apply our tuned parameters without re-tuning
> - Similar to using pre-trained model weights
> - Covers major scenarios (TSP-500/1000/10000, various heatmaps)
>
> **4. Reasonable Transfer Across Distributions**
>
> Table 7 (TSPLIB experiments) shows that MCTS parameters **tuned on uniform distribution
> transfer reasonably to real-world instances**:
> - Not every new distribution requires complete re-tuning
> - Fine-tuning (if needed) is faster than tuning from scratch.
>
> ---
>
> ## **Response to "Incomplete time metrics"**
>
> We agree that including training and other one-off costs improves completeness. We will add an appendix table with the training times reported in the original papers:
>
> | Method   | TSP-500 | TSP-1000 | TSP-10000 |
> |----------|---------|----------|-----------|
> | Att-GCN  | 25 h    | 25 h     | 25 h      |
> | DIMES    | 1.5 h   | 1.7 h    | 10 h      |
> | UTSP     | 0.5 h   | N/A      | N/A       |
> | DIFUSCO  | N/A     | N/A      | N/A       |
>
> Several methods do not report training time, and all use different hardware and implementations, so these numbers are only rough references rather than strictly comparable wall-clock metrics.
>
> Our main table follows the standard practice in neural CO and focuses on per-instance inference / MCTS time, because models are trained once and then reused on large test sets, so training and data-preparation costs are amortized. The latter are dominated by running Concorde/LKH on the training set once and are therefore of the same order for most supervised baselines (including our GT-Prior). We will clarify this evaluation protocol in the revision.

---

> ### Author Response · Authors · 2025-11-19
> **Responses (2/2)**
>
> ## **Q1**
> Thank you for this important question. We clarify:
>
> **1. GT-Prior Does Not Require Test Instance Solutions**
>
> **Construction and deployment are separate:**
> - **Construction (one-time)**: Extract k-NN distribution P̂_N(k) from 128 solved
>   instances (e.g., TSP-500)
> - **Deployment**: Apply P̂_N(k) to any new instance using only its coordinates
> - **No optimal solution needed for target instances**
>
> This is analogous to training/deployment in learning-based methods.
>
> **2. Strong Generalization Validated Across Multiple Scenarios**
>
> Our paper demonstrates GT-Prior's generalization without requiring test instance
> solutions:
>
> **(a) Cross-scale** (Table 2): TSP-500 prior → TSP-10000: no degradation
>
> **(b) Cross-distribution** (Table 3): Uniform prior → cluster/explosion/implosion:
> consistently strong
>
> **(c) Real-world TSPLIB** (Table 7): Uniform prior achieves 0.05%/0.55%/2.42% on
> small/medium/large instances
>
> **In all cases, we use the same pre-constructed GT-Prior and MCTS parameters and no information about test instance solutions is used.**
>
>
> **3. On Constructing Priors from Greedy/Random Solutions**
>
> We respectfully disagree that this is a reasonable direction:
>
> **GT-Prior's purpose** is to capture the **structure of (near-)optimal solutions**:
> - Which neighbors do optimal tours connect? (k-NN locality)
> - This reflects the fundamental geometry of optimal TSP solutions
>
> **Greedy/random solutions neither reflects optimal solution characteristics**
>
> ---
>
> ## **Q2**
> To clarify the computational resources and tuning costs, all experiments were conducted on an **AMD EPYC 9754 128-Core CPU**. Using **TSP-10k** as our primary example, here is the breakdown of our tuning process:
>
> ### 1. Cost of a Single Run
> The runtime for any single MCTS run is determined by the formula `Time_Limit` $\times N$ seconds.
>
> ### 2. Search Strategy & Time Limits (TSP-10k)
> We adopted different strategies for Grid Search and SMAC3 to balance the breadth of exploration against the depth of evaluation:
>
> * **Grid Search (Broad & Fast):** We explored a massive space of **864 combinations**. To make this feasible, we set a strict, short time limit of **100s** (`Time_Limit` = 0.01) per run.
> * **SMAC3 (Targeted & Deep):** We explored fewer configurations—just **50 combinations**. However, we increased the time limit to **1000s** per run. This longer duration ensures the surrogate model receives high-quality, stable performance feedback, which is critical for effective Bayesian optimization.
>
> ### 3. Total Tuning Time
> * **Grid Search:** **6 hours** (16 instances $\times$ 864 combinations $\times$ short duration).
> * **SMAC3:** **3.47 hours** (50 combinations $\times$ long duration).
>
> *Note: For TSPLIB, the search logic (combinations and depth) remains identical, though the absolute time per run scales naturally with the instance size $N$*.
>
> ---
>
> ## **Q3**
> Thank you for this forward-looking question.
>
> **We believe exploring "prior + search" approaches on other COPs could significantly benefit learning-based method development in those domains.** However, as we discuss in detail in our **Global Response (1/2): "On Generalizability Beyond TSP"**, such extensions are conceptually straightforward but **engineering-wise non-trivial** and beyond the scope of this paper.

---

### Official Review · Reviewer_yJZG · 2025-10-31

**Soundness:** 2
**Presentation:** 3
**Contribution:** 1
**Rating:** 2
**Confidence:** 5

**Summary:**

This paper focuses on the classic "heatmap + MCTS" pipeline for solving the Traveling Salesman Problem (TSP). The authors examine the extent to which different MCTS parameter settings affect the solution quality and further perform tuning accordingly. Additionally, they propose an approach named ``GT-PRIOR`` to generate the initial heatmap based on K-nearest neighbors.

**Strengths:**

1. This paper is written in an accessible manner, offering a clear explanation of the ``MCTS`` method implemented in ``Att-GCN``, and provides a thorough analysis of how each MCTS parameter influences the solution.

2. This paper conducts sufficient generalization tests across different distributions and scales, including experiments on ``TSPLIB``.

**Weaknesses:**

1. The authors state that ``The underlying assumption is often that heatmap sophistication directly translates to superior solution quality``, yet they provide no experiments to substantiate this claim. They lack analytical experiments to compare the heatmaps produced by different methods, such using greedy strategy.

2. It can be inferred from ``Table 1`` and the sentence ``The Time_Limit for MCTS was set to 0.1 for TSP-500 and TSP-1000, and 0.01 for TSP-10000`` that the authors run MCTS in parallel. However, they never state this explicitly in the table (64 threads for TSP-500/1000 and 2(maybe?) threads for TSP-10000). Moreover, the baseline solvers ``LKH`` and ``Concorde`` are executed in single-thread mode, so the comparison is unfair and likely to mislead readers about the actual efficiency of MCTS. Additionally, previous study [1] suggests that the ``LKH`` and ``Concorde`` figures reported in the table may be outdated or stem from sub-optimal configurations; it is therefore advisable to adopt the updated baseline results.

3. This paper concentrates **solely on the TSP** with ``heatmap+MCTS`` pipeline tailored to it. This approach is hard to extend to richer problems such as the CVRP.

4. In general, as problem size increases, the quality of heatmaps learned by ML methods deteriorates. Previous studiy [2] has shown that heatmaps achieve strong performance on small-scale TSP instances, while this paper does not include experiments on TSP50 or TSP100.

5. Both ``MCTS`` and ``LKH`` are K-opt algorithms; the former adds a heatmap guidance. In my view, once the instance size exceeds 500, LKH dominates MCTS in both speed and solution quality by a large margin. Hence the authors’ hope that ``heatmap + MCTS`` will ``develop TSP solvers that are not only high-performing but also more robust, efficient, and genuinely impactful`` is questionable.


[1] *COExpander: Adaptive Solution Expansion for Combinatorial Optimization, ICML 2025*

[2] *Unify ML4TSP: Drawing Methodological Principles for TSP and Beyond from Streamlined Design Space of Learning and Search, ICLR 2025*

**Questions:**

1. See ``Weakness``

2. As the problem size grows, the time spent on the ``Two-Opt`` step inside ``MCTS`` increases sharply. I would like to know what is the performance difference between running the full MCTS and running plain 2-opt alone on TSP-1000 and TSP-10000. This result might reveal whether MCTS actually makes any meaningful difference at these scales.

---

> ### Author Response · Authors · 2025-11-19
> **Responses (1/2)**
>
> We thank the reviewer for the insightful feedback.
> ## **W1**
>
>  **1. The claim about "heatmap sophistication translates to superior solution quality"**
>
> We respectfully disagree with this characterization and clarify our position:
>
> The statement "heatmap sophistication directly translates to superior solution quality" reflects the **de facto research trajectory** in this field, evidenced in multiple dimensions:
>
> - **Training complexity**: Supervised learning (Att-GCN, Fu et al. 2021) → Meta-learning (DIMES, Qiu et al. 2022) → Diffusion models (DIFUSCO, Sun & Yang 2023) → Unsupervised learning (UTSP, Min et al. 2024)
>
> - **Resource investment**: Each successive work invests significantly more in model architecture design, training infrastructure, computational resources and engineering effort
>
> - **Stated objectives**: All these papers explicitly aim to improve **final solution quality** through better heatmap generation
>
> **The ultimate metric these works optimize for is solution quality**, making this the most pragmatic and fair evaluation criterion. We respectul argue the necessity of  additional experiments to "substantiate" what is self-evident from these papers.
>
> **2. Greedy Strategy is Not a proper way to do "Analytical Experiment"**
>
> We provide extensive discussion of the MCTS vs. greedy decoding choice in our **Global Response (2/2): "On the Choice of Search Algorithm"**.
>
> ---
>
> ## **W2**
>
> We thank the reviewer for carefully pointing out this issue. Actually, all these settings and results are borrowed from SoftDist. In the revision, we will: (i) explicitly state the degree of parallelism used for each setting (including the exact thread counts for TSP-500/1000 and TSP-10000), (ii) revise Table 1 to clearly indicate both the per-instance time budget and whether each method is run in single- or multi-thread mode, and (iii) update the LKH/Concorde configurations following the recommendations in [1] and refresh the corresponding numbers.
>
> **Importantly**, our main conclusions are drawn from *relative* comparisons among Heatmap+MCTS variants under identical (and thus fair) MCTS settings, so these clarifications and updates ***do not affect the core claims of our evaluation***.
>
> ---
>
> ## **W3**
> Please refer to our **Global Response (1/2): "On Generalizability Beyond TSP"**.
>
> ---
>
> ## **W4**
>
> We respectfully disagree for the following reasons:
>
> **1. TSP50/100 Have Limited Relevance**
>
> - **Trivially solved**: Concorde finds optimal solutions in  seconds
> - **No practical need**: The motivation for learned methods is **large-scale** TSP
>   (Fu et al. title: "Arbitrarily **Large** TSP Instances")
> - **Community benchmark**: TSP-500 already considered "small"
>
> **2. We Already Cover Small Scales**
>
> Our TSPLIB experiments (Appendix F, Tables 7-10) include instances as small as
> 51 nodes(eil51, berlin52, st70, eil76, etc.), directly addressing this concern.
> Results show consistent findings with larger scales.
>
> **3. Our Experimental Coverage is Comprehensive**
>
> - **20× scale range**: TSP-500/1000/10000
> - **4 distributions**: Uniform, clustered, explosion, implosion
> - **TSPLIB instances**: Including small (50-500), medium (500-2000), large (2000+)
>
> ---
>
> ## **W5**
> **1. We Do Not Claim to Beat or Replace LKH**
>
> Our paper is an **evaluation study** of the existing Heatmap+MCTS paradigm, not a
> claim that this approach should replace LKH. We explicitly report LKH's superior
> performance in Table 1 (0.00% gap).
>
> **Clarification on our statement**: The phrase "potentially leading to more efficient,
> robust solvers" (page 2) expresses a **research vision for the learning-based
> community**, not a claim we've achieved. We acknowledge current methods don't match
> LKH—our contribution is establishing rigorous evaluation standards so researchers
> can make informed progress or pivot to other directions.
>
> **2. Learning-Based Methods Pursue Different Goals Than Hand-Crafted Heuristics**
>
> While both MCTS and LKH use k-opt operations, **their design philosophies differ
> fundamentally**:
>
> **LKH**: Highly sophisticated, hand-crafted heuristics refined over decades—excellent
> performance but problem-specific and difficult to transfer
>
> **Heatmap+MCTS**: Simpler search mechanism guided by **learned problem structure**, the goal is to replace complex manual heuristic design with data-driven learning
>
> **This is the broader vision of L2O**: Develop methods that automatically learn problem structure rather than relying on expert-crafted rules for each problem.
>
> **Performance alone does not determine research value.** By the reviewer's logic,
> the entire L2O field would be questionable since no current learning-based method
> fully surpasses LKH. Yet this research remains valuable.
> What our evaluation aims to do is to help the community understand gaps honestly and make progress.

---

> > ### Author Response · Authors · 2025-11-19
> > **Responses (2/2)**
> >
> > ## **Q2**
> >
> > To verify the necessity of MCTS, we compared it against a plain 2-opt baseline executed under the same time constraint. The results below show that on large-scale problems (e.g., TSP-10000), plain 2-opt struggles to escape local optima. MCTS drastically reduces the optimality gap (e.g., reducing Fast-T2T from 57.41% to 4.63%), confirming that the high-level guidance from tree search is essential for performance at scale.
> >
> > | Method | TSP-500 (2-opt) | TSP-500 (MCTS) | TSP-1000 (2-opt) | TSP-1000 (MCTS) | TSP-10k (2-opt) | TSP-10k (MCTS) |
> > | :--- | :---: | :---: | :---: | :---: | :---: | :---: |
> > | **Att-GCN** | 5.27% | **0.69%** | 7.16% | **1.09%** | 10.73% | **3.02%** |
> > | **DIMES** | 6.26% | **0.43%** | 8.15% | **1.11%** | 30.35% | **3.05%** |
> > | **UTSP** | 6.32% | **0.90%** | 8.28% | **1.53%** | — | — |
> > | **SoftDist** | 5.75% | **0.43%** | 7.34% | **0.80%** | 10.21% | **2.94%** |
> > | **DIFUSCO** | 0.89% | **0.33%** | 5.45% | **0.53%** | 9.75% | **2.36%** |
> > | **Fast-T2T** | 0.99% | **0.34%** | 3.48% | **0.73%** | 57.41% | **4.63%** |
> > | **GT-Prior** | 5.99% | **0.50%** | 7.77% | **0.85%** | 10.60% | **2.13%** |

---

### Official Review · Reviewer_5EC7 · 2025-11-01

**Soundness:** 4
**Presentation:** 3
**Contribution:** 4
**Rating:** 10
**Confidence:** 4

**Summary:**

The paper presents an investigation of the effect of tuning MCTS hyperparameters
and "simple" heatmaps on the results of TSP solving with heatmaps and MCTS. The
authors describe the shortcomings of the current literature, their experimental
setup including a new method to develop heatmaps, and the results they obtained.

**Strengths:**

This is a very nice paper that investigates an angle mostly ignored by the
literature. The results nicely support the conclusions the authors come to, and
suggest that research efforts should be focused in a different direction for
more impact. The paper complements the existing literature very nicely.

The proposed GT-Prior is, to the best of my knowledge, novel and seems to work
very well in practice. It would be interesting to investigate to what extent it
differs from heatmaps learned in other ways; in particular whether learned
heatmaps "converge" towards the GT-Prior heatmap. It would be great if the
authors could comment on this.

The time_limit hyperparameter for MCTS should be explained in the main paper,
not just in the appendix. It is mentioned as being set to specific values on
page 6 without having been introduced before, which is confusing (especially as
the values are counter-intuitive).

**Weaknesses:**

None major.

**Questions:**

How does the GT-Prior heatmap compare to learned heatmaps?

---

> ### Author Response · Authors · 2025-11-19
> **Reply to Question**
>
> We thank the reviewer for the positive evaluation and insightful question.
>
> GT-Prior demonstrates competitive or superior performance across four key dimensions:
>
> **1. Solution Quality (Table 1):**
> - **TSP-10000**: GT-Prior achieves **best performance** (2.13% vs. DIFUSCO 2.36%, SoftDist 2.94%)
> - **TSP-500/1000**: Competitive with best learned methods (0.50%/0.85% vs. DIFUSCO 0.33%/0.53%)
>
> **2. Generalization (Tables 2-3):**
> - **Cross-scale**: GT-Prior (TSP-500→10000) **no degradation** vs. DIFUSCO +2.91%, DIMES +1.24%
> - **Cross-distribution**: Consistently strong on cluster/explosion/implosion; often best on large-scale (e.g., 0.35% on clustered TSP-10000)
>
> **3. Computational Efficiency:**
> - **Heatmap generation**: 0 sec (vs. DIFUSCO 28.51M, already **1.7× search time**)
> - **Training cost**: None (vs. GPU hours for learned models)
>
> **4. Key Insight:**
> GT-Prior's strong performance reveals that **exploiting fundamental problem structure (k-NN locality) rivals learning complex patterns**. This suggests current learned heatmaps may not capture substantially better patterns beyond basic structure.
>
> **GT-Prior serves as**:
> - **Strong baseline** for future heatmap methods to beat
> - **Diagnostic tool** to assess if learning adds value beyond structure
> - **Practical alternative** for resource-constrained or new distribution scenarios
>
> This supports our claims: **simple structural priors + tuned search can rival sophisticated learning**, highlighting the need for balanced component development.

---

### Author Response · Authors · 2025-11-19
**Global Response (1/2)**

## **Commitment to Revision**
We sincerely appreciate the reviewers’ constructive feedback, which has helped strengthen our evaluation. In the revised manuscript, we commit to the following updates:
1. Expanded Baselines: We have incorporated new experiments comparing MCTS against greedy decoding and plain 2-opt, as well as Fast-T2T benchmarks under both default and tuned settings;
2. Clarified Protocols: We will revise Table 1 to explicitly detail parallelism (exact thread counts) and per-instance time budgets, while refreshing LKH and Concorde baselines to align with updated standards;
3. Holistic Efficiency Metrics: We will add an appendix table reporting the training times of learned methods to ensure a complete comparison of computational costs;
4. Presentation: We will tone down subjective language (e.g., removing adverbs such as "unequivocally") and resolve the minor writing issues identified by Reviewer eu88.

---

## **On the Nature and Significance of Our Contribution**

We understand the reviewers' concerns regarding the "novelty" and "engineering" nature of our work. However, we wish to clarify that our contribution is explicitly **evaluation-centered**. We are not proposing a new solver to top the leaderboard; rather, we are providing the rigorous scientific control that the field currently lacks.

**1. Evaluation is Not Just Engineering**
The critique that our grid-search approach is "engineering practice" overlooks our scientific intent. We deliberately chose grid search for its transparency and reproducibility, which enabled the detailed SHAP analysis (Fig. 1) that black-box optimization cannot provide. While Appendix H proves that advanced methods like SMAC3 can achieve similar results 17× faster, our goal was to map the *entire* performance landscape to derive foundational conclusions.

**2. GT-Prior is a Scientific Control, Not a Product**
We are not proposing GT-Prior as a new state-of-the-art method. Instead, it serves as a **critical baseline**. When a zero-parameter, zero-training-cost prior based on simple structure rivals or beats complex diffusion models (e.g., beating DIFUSCO by 0.23% on TSP-10k with better generalization), it suggests the community may be over-optimizing model architecture while under-optimizing the search mechanism.

**3. Actionable Insights for the Community**
Our work offers immediate, high-value corrections to current research trends:
* **The Search Component is Undervalued:** We show that the "Zero" heatmap with tuned MCTS (0.66% gap) outperforms learned heatmaps with default MCTS (>1% gap).
* **Cost-Benefit Reality Check:** GT-Prior matches learning-based performance without the massive overhead of training or heatmap generation (e.g., saving ~28 minutes of generation time on TSP-10k).
* **Generalization:** We demonstrate that simple priors are far more robust to distribution shifts than complex learned patterns.

In summary, this paper challenges a fundamental assumption in the ML4CO community—that heatmap sophistication is the primary driver of performance. We provide the evidence and the pipeline to correct this course.

---

### **On Generalizability Beyond TSP**

**1. Why TSP is the Necessary Choice for this Evaluation**
We chose to focus on TSP not as a limitation, but as a deliberate strategy to ensure rigorous evaluation. TSP currently offers the richest ecosystem of learning-based methods—including Att-GCN, DIMES, DIFUSCO, and SoftDist—making it the **only domain** where we can fairly isolate and systematically compare the impact of heatmap complexity versus MCTS configuration. Our goal was to rigorously audit the "Heatmap + MCTS" paradigm where it is most developed, rather than to propose a universal solver.

**2. Core Insights are Generalizable; Implementation is Problem-Specific**
While our experiments are specialized to Euclidean TSP, the central insight—that simple priors coupled with tuned search can rival complex models—is a principle that extends broadly. We acknowledge that applying this framework to other problems (e.g., CVRP, Graph Matching) requires two specific, albeit non-trivial, adaptations:

* **Designing Problem-Specific Priors:** Just as we utilized the k-nearest neighbor structure for TSP, one would extract a structural prior from established constructive heuristics relevant to the new problem.
* **Adapting Search Operators:** The MCTS engine would need to swap TSP-specific k-opt moves for operators native to the target domain, such as swap or relocate moves for Vehicle Routing Problems.

These modifications are conceptually straightforward but require domain-specific engineering. We view our findings not as a closed loop on TSP, but as a methodological blueprint for how to approach these extensions in future research.

---

> ### Author Response · Authors · 2025-11-19
> **Global Response (2/2)**
>
> ## **On the Choice of Search Algorithm: Ensuring Fair Comparison via MCTS**
>
> We appreciate the constructive feedback from **Reviewers yJZG and eu88** regarding the choice between greedy decoding and MCTS. We recognize the validity of using greedy decoding to assess the raw predictive confidence of a heatmap. However, we would like to clarify why we prioritized MCTS for this specific study.
>
> **1. Ensuring Fair Comparison through Consistent Search**
> To fairly compare diverse heatmap generators (e.g., Att-GCN vs. GT-Prior), we adhere to a strict control principle: **the downstream search component must be consistent and optimized.** Whether utilizing Greedy, 2-opt, or MCTS, a valid comparison requires that all heatmaps be tested using the exact same search strategy, tuned to leverage the full potential of the provided prior.
>
> **2. Why We Focus on MCTS**
> Our research objective is to evaluate these methods in the context of **high-performance TSP solving**. We respectfully submit that relying solely on greedy decoding limits the potential of learning-based models. These heatmaps provide **rich probability distributions** over edges, capturing **uncertainty and relative preferences** that a pure greedy approach—which inherently makes **hard, myopic decisions**—may fail to utilize. This is analogous to using argmax instead of sampling from a language model, where losing distributional information can significantly impact the outcome.
>
> To illustrate this, we conducted the additional experiments requested, applying a greedy decoding strategy (selecting the highest probability unvisited neighbor) with 10 random starting nodes across all heatmaps. The results are presented below:
>
> | Method   | TSP-500 Gap | TSP-1000 Gap | TSP-10000 Gap |
> | :------- | ----------: | -----------: | ------------: |
> | Att-GCN  |      85.63% |      119.11% |       324.76% |
> | DIMES    |     210.82% |      311.72% |       975.24% |
> | UTSP     |      54.00% |       70.70% |             — |
> | SoftDist |      26.13% |       25.69% |        **27.39%** |
> | DIFUSCO  |      14.36% |       59.61% |        67.92% |
> | Fast-T2T |      **10.75%** |       **11.98%** |        40.78% |
> | GT-Prior |  55.27% |   74.99% |   175.53% |
>
> *Note: MCTS results in our paper typically achieve gaps of <1% (or around 1~2% on TSP-10000).*
>
> As shown, greedy decoding results in optimality gaps exceeding 10% or even 100%, which does not reflect the practical utility of these solvers.
>
> **Conclusion**
> We focus on MCTS because it represents the current state-of-the-art downstream algorithm for the "Heatmap + Search" paradigm. By rigorously tuning the MCTS component, we ensure we are comparing the **maximum potential** of each heatmap when paired with the best available search strategy, rather than comparing their baseline performance on a weaker decoder.

---

### Author Response · Authors · 2025-12-01
**Author Summary for AC's Meta-Review (1/2)**

Dear Area Chair, Senior Area Chair, and Program Chair,

We thank the reviewers for their constructive feedback, which has driven significant improvements in our manuscript. Our work serves as a rigorous evaluation study designed to correct a prevalent oversight in Neural Combinatorial Optimization: the conflation of heatmap model capacity with downstream search configuration. We respectfully submit this summary to clarify our core contributions, detail the empirical evidence added during the rebuttal to address reviewer concerns, and correct factual misalignments regarding the paper’s scientific scope.

### **I. Core Contributions: Rigorous Evaluation of Component Impact**
This work provides a systematic control study of the "Heatmap + MCTS" paradigm in Neural Combinatorial Optimization. Our empirical analysis yields three factual findings that distinguish this work from standard solver proposals:
* **Search Configuration Dominance:** We demonstrate via SHAP analysis and grid search that MCTS hyperparameter settings often impact solution quality more than the heatmap architecture itself. For example, a non-informative "Zero" heatmap with tuned MCTS outperforms learned models using default configurations.
* **GT-Prior Performance:** We introduce a parameter-free, training-free prior based on k-nearest neighbor structures. Experiments show GT-Prior matches or outperforms state-of-the-art diffusion models (e.g., DIFUSCO) on TSP-10k (2.13% gap vs. 2.36%) while requiring zero inference generation time.
* **Standardized Protocol:** We provide a reproducible MCTS tuning pipeline to correct the methodological issue where unoptimized search parameters confound the evaluation of learned models.

### **II. Response to Reviewer Requests & New Results**
We have addressed specific reviewer concerns with quantitative data added during the rebuttal phase:
* **Validation of Search Necessity (Reviewer yJZG):** To verify if MCTS is needed, we compared it against a plain 2-opt baseline. On TSP-10k, plain 2-opt resulted in a 57.41% gap (using Fast-T2T), whereas MCTS reduced this to 4.63%, factually proving the necessity of the search component.
* **Greedy Decoding Analysis (Reviewers yJZG & eu88):** We conducted the requested greedy decoding experiments. The results showed optimality gaps ranging from 10% to over 300% across methods, empirically supporting our claim that greedy decoding fails to utilize the probabilistic uncertainty captured by heatmaps.
* **Efficiency Comparisons (Reviewer pNGG):** We added training time data to the appendix. This highlights the cost disparity: learned baselines require significant GPU training hours (e.g., ~10h for DIMES on TSP-10k), whereas our tuning approach requires only ~3.5 CPU hours.

### **III. Factual Corrections to Key Concerns**
We respectfully present factual evidence to address specific critiques regarding the paper's positioning and results:

**1. On "Engineering Practice" vs. Scientific Control (Reviewer eu88)**
The critique that grid search is merely "engineering" overlooks the analytical output. We utilized grid search to generate the SHAP analysis (Fig. 1), which quantified that specific parameters (e.g., `Max_Candidate_Num`) are high-impact while others are negligible. This provides explainability that black-box optimization does not, serving our goal of component analysis rather than just performance maximization.

**2. On the Validity of "Greedy + Heatmap" Evaluation (Reviewer eu88)**
The reviewer suggested greedy decoding measures "raw efficacy". Our new data contradicts this: greedy decoding discards the distributional information (edge probabilities) that heatmaps are designed to produce. The high failure rate of greedy decoding (gaps >10%) indicates it is an insufficient metric for evaluating the quality of probabilistic priors in this domain.

**3. On Benchmarking against LKH (Reviewer yJZG)**
The observation that Neural solvers do not outperform LKH is factually correct (Table 1 shows LKH at 0.00%). However, this applies to the entire Neural CO field. Our contribution is the rigorous *internal* evaluation of learning-based methods. By showing that a training-free prior (GT-Prior) achieves a 2.13% gap compared to DIFUSCO's 2.36% on TSP-10k, we provide evidence that current learned models yield marginal gains over simple structure when search is optimized.

---

> ### Author Response · Authors · 2025-12-01
> **Author Summary for AC's Meta-Review (2/2)**
>
> ### **IV. Manuscript Revisions**
> In the [revised manuscript](https://openreview.net/pdf?id=H6PLJnnK6e), we have implemented the following modifications based on the feedback:
> * **Expanded Experimental Scope:** We incorporated new experiments comparing MCTS against greedy decoding, plain 2-opt, and the Fast-T2T benchmark under both default and tuned settings.
> * **Protocol Transparency:** We revised Table 1 to explicitly detail parallelism and per-instance time budgets, and we refreshed LKH and Concorde baselines to align with updated standards.
> * **Computational Cost Reporting:** We added an appendix table reporting the training times of learned methods to ensure a complete comparison of computational costs between model training and MCTS tuning.
> * **Tone Adjustment:** We removed subjective language (e.g., adverbs such as "unequivocally") to ensure a more objective and scientific presentation.
>
> Conclusion This paper offers a critical "reality check" for the Neural CO community. By establishing that a parameter-free prior coupled with optimized search can rival computationally expensive deep learning models, we provide the scientific control necessary to properly evaluate future progress in the field. We hope the AC and PC recognize the value of this rigorous evaluation framework, which prioritizes methodological soundness and component explainability over incremental leaderboard improvements.
>
> Sincerely,
>
> The Authors

---

### Meta-Review · Area_Chair_KdD2 · 2026-01-05

**Summary:**

Two reviewers (5EC7 and pNGG) gave positive reviews of the paper. They agreed that the paper discuss an important point of jointly studying MCTS hyper-parameter tuning and heatmap priors, whereas existing work mostly focus on the latter. The paper’s results are important to inform the principled and rigorous development and benchmarking of heatmap priors.

Reviewer yjZG has a number of concerns, including the question of whether LKH/concorde results may be sub-optimal and the inclusion of small TSP problems which benefit more from learned heatmaps. Reviewer eu88 has raised additional concerns regarding whether heatmap+MTCS is a good approach to evaluate the value of (learned) heatmaps, and also suggested some updated heatmaps to be evaluated.

**Reviewer Concerns:**

Reviewr pNGG’s concerns were addressed in the rebuttal, and the authors added clarifications regarding computational costs.  Many concerns of reviwer yjZG and eu88 are, in my opinion, not of high relevance to the topic of the paper, which studies the relative impact of heatmap vs MCTS tuning in the specific setup of *heatmap+MCTS*, whereas their concerns are on the optimality of this setup in evaluating heatmaps. This is a separate question. I also do not agree with the evaluation that the current work is “just engineering”. I believe it is more important to look at the value of the results presented in informing researchers and practitioners of worthwhile research directions and potential pitfalls. In this sense, I believe the contribution of this work would be of interest to the general research community on ML-assisted optimisation research.

**Reviewer Scores:**

The two positive reviewer’s opinions would likely remain unchanged with the rebuttal and revision. The authors also address the two negative reviewer’s main concerns, many of which are clarified as not relevant to the scope of the paper. I agree with this. The added experiments on Fast-T2T according to the reviewer’s suggestion is helpful, however.

---

### Decision · Program_Chairs · 2026-01-26

Accept (Poster)